# Synergistic Effects of UVB and Ionizing Radiation on Human Non-Malignant Cells: Implications for Ozone Depletion and Secondary Cosmic Radiation Exposure

**DOI:** 10.3390/biom15040536

**Published:** 2025-04-06

**Authors:** Angeliki Gkikoudi, Gina Manda, Christina Beinke, Ulrich Giesen, Amer Al-Qaaod, Elena-Mihaela Dragnea, Maria Dobre, Ionela Victoria Neagoe, Traimate Sangsuwan, Siamak Haghdoost, Spyridon N. Vasilopoulos, Sotiria Triantopoulou, Anna Georgakopoulou, Ioanna Tremi, Paraskevi N. Koutsoudaki, Sophia Havaki, Vassilis G. Gorgoulis, Michael Kokkoris, Faton Krasniqi, Georgia I. Terzoudi, Alexandros G. Georgakilas

**Affiliations:** 1DNA Damage Laboratory, Physics Department, School of Applied Mathematical and Physical Sciences, National Technical University of Athens (NTUA), Zografou Campus, 15780 Athens, Greece; angelikigkikoudi@mail.ntua.gr (A.G.); svasilopoulos@mail.ntua.gr (S.N.V.); ge19141@mail.ntua.gr (A.G.); ioannatre@med.uoa.gr (I.T.); 2Laboratory of Health Physics, Radiobiology & Cytogenetics, Institute of Nuclear & Radiological Sciences & Technology, Energy & Safety, National Centre for Scientific Research “Demokritos”, 15341 Agia Paraskevi, Greece; iro@rrp.demokritos.gr (S.T.); gterzoudi@rrp.demokritos.gr (G.I.T.); 3Radiobiology Laboratory, “Victor Babeș” National Institute of Pathology, 99-101 Splaiul Independentei, 050096 Bucharest, Romania; gina.manda@ivb.ro (G.M.); elenamihaeladragnea@gmail.com (E.-M.D.); maria.dobre@ivb.ro (M.D.); ionela.neagoe@ivb.ro (I.V.N.); 4Bundeswehr Institute of Radiobiology, University of Ulm, Neuherbergstraβe 11, 80937 Munich, Germany; christinabeinke@bundeswehr.org; 5Physikalisch-Technische Bundesanstalt (PTB), Bundesallee 100, 38116 Braunschweig, Germanyamer.al-qaaod.ext@ptb.de (A.A.-Q.); faton.krasniqi@ptb.de (F.K.); 6ABTE/ToxEMAC Laboratory, University of Caen Normandy, F-14050 Caen, France; traimate.sangsuwan@su.se (T.S.); siamak.haghdoost@unicaen.fr (S.H.); 7Department of Molecular Biosciences, The Wenner-Gren Institute, Stockholm University, SE-10691 Stockholm, Sweden; 8Molecular Carcinogenesis Group, Department of Histology and Embryology, Medical School, National and Kapodistrian University of Athens, 11527 Athens, Greece; pkoutsoudaki@med.uoa.gr (P.N.K.); shavaki@med.uoa.gr (S.H.); vgorg@med.uoa.gr (V.G.G.); 9Biomedical Research Foundation, Academy of Athens, 11527 Athens, Greece; 10Ninewells Hospital and Medical School, University of Dundee, Dundee DD2 1SG, UK; 11Faculty Institute for Cancer Sciences, Manchester Academic Health Sciences Centre, University of Manchester, Manchester M20 4GJ, UK; 12Faculty of Health and Medical Sciences, University of Surrey, Guildford GU2 7YH, UK; 13Group of Nuclear Physics, Physics Department, School of Applied Mathematical and Physical Sciences, National Technical University of Athens (NTUA), 15780 Zografou, Greece; kokkoris@central.ntua.gr

**Keywords:** solar UV radiation, UVB, secondary cosmic radiation, ozone, biological effects, DNA damage, fibroblasts, keratinocytes, monocytes

## Abstract

The ozone layer in the Earth’s atmosphere filters solar radiation and limits the unwanted effects on humans. A depletion of this ozone shield would permit hazardous levels of UV solar radiation, especially in the UVB range, to bombard Earth’s surface, resulting in potentially significant effects on human health. The concern for these adverse effects intensifies if we consider that the UVB solar radiation is combined with secondary cosmic radiation (SCR) components, such as protons and muons, as well as terrestrial gamma rays. This research aims to delve into the intricate interplay between cosmic and solar radiation on earth at the cellular level, focusing on their synergistic effects on human cell biology. Through a multidisciplinary approach integrating radiobiology and physics, we aim to explore key aspects of biological responses, including cell viability, DNA damage, stress gene expression, and finally, genomic instability. To assess the impact of the combined exposure, normal i.e., non-malignant human cells (skin fibroblasts, keratinocytes, monocytes, and lymphocytes) were exposed to high-energy protons or gamma rays in combination with UVB. Cellular molecular and cytogenetic biomarkers of radiation exposure, such as DNA damage (γH2AΧ histone protein and dicentric chromosomes), as well as the expression pattern of various stress genes, were analyzed. In parallel, the MTS reduction and lactate dehydrogenase assays were used as indicators of cell viability, proliferation, and cytotoxicity. Results reveal remaining DNA damage for the co-exposed samples compared to samples exposed to only one type of radiation in all types of cells, accompanied by increased genomic instability and distinct stress gene expression patterns detected at 24–48 h post-exposure. Understanding the impact of combined radiation exposures is crucial for assessing the health risks posed to humans if the ozone layer is partially depleted, with structural and functional damages inflicted by combined cosmic and UVB exposure.

## 1. Introduction

The influx of high-energy particles from space into the Earth’s atmosphere can lead to a variety of atmospheric processes. These events generate additional charged particles, ions, and free electrons in the upper atmosphere, which can influence both the electrical conductivity of the atmosphere and its chemical composition [1]. In particular, during solar particle events, these high-energy particles can lead to the production of chemically active nitrogen (NO_x_) and hydrogen oxides (HO_x_), which are known to be associated with the catalytic breakdown of ozone (O_3_). As ozone protects the Earth from harmful ultraviolet (UV) rays from the Sun, its depletion would lead to an increase in biologically active UV radiation flux, with significant impact on the exposed organisms [2]. Ionizing radiation (IR) is a potent genotoxic agent that induces DNA damage, including clustered DNA lesions, a significant contributor to the mutagenic and carcinogenic potential of IR [3,4,5]. UVB radiation (280–315 nm), in contrast, does not directly induce the formation of double-strand breaks (DSBs). Even thought is particularly concerning, as it directly damages DNA by inducing cyclobutane pyrimidine dimers (CPDs) and pyrimidine (6-4) pyrimidone photoproducts (6-4 PPs), leading to replication fork stalling. Additionally, UVB generates reactive oxygen species (ROS), further contributing to oxidative DNA damage. While nucleotide excision repair (NER) is the primary mechanism for UV-induced lesions, its efficiency varies, and unrepaired damage can lead to DSBs during replication [6,7,8,9,10,11,12,13,14,15,16,17,18].

UVB radiation has been identified as a primary factor causing various skin conditions such as erythema and oedema [19]. The minimal erythema dose (MED) for UVB radiation is between 300 J/m^2^ and 800 J/m^2^. In contrast, the MED for UVA radiation can be more than two orders of magnitude higher [20]. Exposure to these types of UV radiation, e.g., due to a depletion of stratospheric ozone, is associated with an increased risk of non-melanoma and melanoma skin cancers, potential changes in immune function, and the development of cataracts. Additionally, photoaging is primarily caused by UVB radiation, although UVA radiation can also contribute to this effect [21]. To illustrate the effect of atmosphere in the UVB part of the solar spectrum, Appendix A shows the reference AM 1.5 spectra defined by the American Society for Testing and Materials (ASTM) http://rredc.nrel.gov/solar/spectra/am1.5/, accessed on 1 February 2025). The figure also shows the difference between the extraterrestrial solar spectrum and the direct spectrum (the direct spectrum considers only the direct component of sunlight, ignoring any diffuse radiation), which allows us to see the effect of the atmosphere, including ozone, in the UVB part of the spectrum (Appendix A). The total extraterrestrial UVB flux density blocked by the atmosphere is about 17 W/m^2^. Ozone depletion therefore causes some of this energy to leak into the ground level.

In parallel, cosmic ray protons at ground level exhibit a broad and relatively flat energy distribution spanning from 0.01 MeV to 100 MeV, as shown in Figure 1 (dashed red line) [22]. Figure 1 compares the total stopping power of protons and muons in A-150 tissue-equivalent plastic, which is used to mimic the properties of human tissue in radiation fields. Secondary cosmic proton and muon spectra at an altitude of 75 m and a cut-off rigidity of 2.89 GV is as well shown in Figure 1, as simulated with the EXPACS program [23]. The cosmic flux increases exponentially with altitude (up to an altitude of approx. 15 km) and can be accompanied by additional increases during ground level enhancements [24]. Beyond 100 MeV, the proton flux drops sharply. The 10 MeV energy used in our experiments aligns with the broad peak centered around 45 MeV. While the proton flux is typically low under normal solar conditions at ground level (~10^−6^ cm^−2^ s^−1^ MeV^−1^), it can increase significantly during major solar particle events or at higher latitudes. For example, the proton flux rises by over two orders of magnitude when moving from ground level to an altitude of approximately 10 km (in the range of cruising altitudes for commercial airliners). For additional information, see Appendix A.

The combination of secondary cosmic radiation and solar UVB exposure, due to ozone depletion, creates a complex radiation environment that may present unique challenges to biological systems. Despite the significance of such exposures, there is a notable gap in the literature regarding the synergistic effects of ionizing radiation and UVB on human cells. The aim of this study is to address this gap by systematically investigating the biological consequences of combined radiation exposures, providing insights into potential risks. Human skin and blood cell lines, as well as in primary human lymphocytes, were exposed in simulated extreme exposures that partially mimic the impact of large solar particle events combined with ozone depletion and consequent ground-level enrichment. Irradiation facilities provided radiation fields with a similar radiation weighting factor as muons (Co-60 and Cs-137 gamma radiation) and similar stopping power (MeV protons), and calibrated UV radiation sources simulated the UVB exposure in the atmosphere. The cosmic radiation field was partially simulated with 10 MeV protons, which have a total stopping power of about 5 keV/µm in tissue-equivalent material, similar to muons at the lower end of the energy spectrum [22]. This work was performed in the framework of the European Partnership on Metrology (EPM) joint research project BIOSPHERE, which aims to develop the necessary instrumentation, methods, and measurement infrastructure to assess how the increasing ionization of the atmosphere, caused by extraterrestrial radiation fields (cosmic rays and solar UV radiation) amplified by anthropogenic emissions, affects human health (https://euramet-biosphere.eu/, accessed on 1 February 2025). Based on the results presented here, this project will provide data to evaluate the contribution of cosmic rays and UV radiation exposures in the risk for developing chronic diseases and cancer under extreme conditions such as ozone-depleted atmosphere with ground level enhancements (e.g., during strong solar particle events).

## 2. Materials and Methods

### 2.1. Cells

#### 2.1.1. Human Non-Cancerous Cell Lines

Human HaCaT keratinocytes were purchased from CLS Cell Line Services GmbH (Eppelheim, Germany), while Hs27 skin fibroblasts (CRL-1634) and human peripheral blood monocytes (CRL 9855, US Pat. No. 5.447.861) were obtained from the American Tissue and Cell Collection (ATCC, Manassas, VA, USA), and were cultivated according to the recommendations of the depositor. Thus, the adherent cell lines HaCaT and Hs27 were cultivated in Dulbecco’s Modified Eagle’s Medium (DMEM, 12491015, Life Technologies, Burlington, ON, Canada) supplemented with 10% fetal bovine serum (FBS, A3160401, Life Technologies, Burlington, ON, Canada), with passage 2–3 times per week using Trypsin/EDTA 0.05%/0.02% in PBS, w/o Ca^2+^ and Mg^2+^ (P10-023100, PAN Biotech GmbH, Aidenbach, Germany). The non-adherent CRL 9855 monocytes were grown in Iscove’s Modified Dulbecco’s Medium (12440053, IMDM, Life Technologies, Burlington, ON, Canada) containing 10% FBS, 1% HT supplement (10 mM sodium hypoxanthine and 1.6 mM thymidine) (11-067-030, Gibco, Life Technologies, Burlington, ON, Canada), 1% 2-mercaptoethanol (21985023, Gibco, Life Technologies, Burlington, ON, Canada), and 1% antibiotic-antimycotic solution (100×) (A5955, Sigma-Aldrich, Saint Louis, MO, USA). Cells were passaged 2–3 times per week by addition of fresh complete culture medium for keeping cellular density between 0.35 × 10^6^ cells/mL and 1.30 × 10^6^ cells/mL.

#### 2.1.2. Human Primary Lymphocytes (PBMCs)

Two apparently healthy probands (1 female, 1 male) were enrolled in this study with informed consent according to the approval of the Institutional Review Board at the University of Ulm (no. 410/22). Peripheral blood samples were collected in lithium-heparinized CPT vials (BD #362780, Heidelberg, Germany), and lymphocytes were isolated according to the manufacturer’s manual.

### 2.2. Preparation of Cells for Exposure to Protons and UVB

#### 2.2.1. Human Cell Lines

Cells at passages 8-16, with a viability > 90%, were used in experiments. Briefly, adherent HaCaT and Hs27 cells were detached with Trypsin/EDTA 0.05%/0.02% (P10-023100, PAN Biotech GmbH, Aidenbach, Germany). All the cell suspensions (HaCaT, Hs27 and CRL 9855) were centrifuged and suspended in RPMI 1640 culture medium without phenol red (11835030, Gibco, Life Technologies, Burlington, ON, Canada), supplemented with 10% FBS. Cells were counted using a Burker-Turk counting chamber, and viability was assessed by the trypan blue exclusion test. Cells were suspended in RPMI 1640 culture medium without phenol red at a cell density of 0.7 × 10^7^ cells/mL for HaCaT or Hs27 cells, and 10^7^ cells/mL for CRL 9855 monocytes. Finally, 100 µL of cell suspension was placed in a well of a low-adhesion 48-well plate (677102, Cellstar, Greiner Bio One, Kremsmünster, Austria) or in the dish described in Appendix A), for proton exposure top-to-bottom or bottom-to-top, respectively. Distinct samples were prepared for UVB or proton exposure, as well as samples for the co-exposure of cells to protons and UVB. Various non-irradiated controls, containing cells that were handled the same as the irradiated ones, were used. The sham-irradiated cells (IrrCTRL) were kept in the irradiation room, outside the proton beam, while the lab control (CTRL) was kept in the biology laboratory, placed outside the irradiation room. Co-exposed samples were first irradiated with protons and, within approx. 20 min thereafter, exposed to UVB. Cells exposed first to UVB and then to protons were investigated as well. Cells were placed in clear low-adhesion 48-well plates for UVB exposure. Thus, cells exposed to protons in low-adhesion 48-well plates were directly transferred to UVB exposure, while cells placed in dishes for proton exposure were first transferred to low-adhesion 48-well plates, and were exposed to UVB thereafter (see also Appendix A).

#### 2.2.2. PBMCs

Irradiations were performed in three different days in 48-wellplates (CELLSTAR^®^ #677180, Greiner, Frickenhausen, Germany). For each condition, 5 × 10^6^ lymphocytes suspended in 100 µL cell RPMI culture medium without phenol red (# 11835030, Gibco, Germany) were radiation-exposed or sham-irradiated at room temperature (RT). To determine the dose–response relationship of proton-induced dicentric formation, lymphocytes from both donors were exposed to 0.1 Gy, 0.25 Gy, 0.5 Gy, 0.75 Gy, and 1.0 Gy protons (10 MeV, 4.7 keV/µm for cells located on well ground). Each proton exposure was accompanied by a plate control (100 µL cell suspension placed in a well on the same well plate, but not exposed to the proton beam) to determine the dicentric yield induced by the neutron–gamma background which originates from the proton beam generation. This background radiation was measured to be 7 mGy per 0.5 Gy proton dose on irradiation day two and three. For analysis of combined exposures, cells were exposed to 0.5 Gy protons (10 MeV, ~4.7 keV/µm for cells located on well ground) or 400 J/m^2^ or 100 J/m^2^ UVB, respectively, and vice versa. Two irradiation experiments with blood from donor one were performed for proton, UVB, and combined exposures by setting up replicate cultures (four cultures in total per condition). The 0.5 Gy proton exposures of cells were accompanied by a plate control. Additionally, the baseline dicentric frequency in unexposed lymphocytes (remaining in the laboratory) was determined (PTB lab control). After irradiation, 900 µL RPMI (#11875093, Gibco, Germany) supplemented with 20% fetal calf serum and 1% penicillin/streptomycin were added to irradiated cells, samples were split into two cultures (500 µL culture volume/well) and kept for at least 2 h at 37 °C for DNA repair.

### 2.3. Preparation of Cells for Exposure to Gamma Rays and UVB

#### Human Non-Cancerous Cell Lines

The Hs27, VH10, and HaCaT cells were plated on 22 × 22 mm^2^ coverslips in 35 mm dishes and incubated overnight. The next day, they were exposed to 0.5 Gy gamma rays, 100 J/m^2^ of UVB, or a combination of both radiations. After the exposure, cells were incubated at 37 °C for the desired time points (1, 4 h, and 24 h for the Hs27 cell line, 1, 5 h, and 24 h for the VH10 cell line, and 1 h, 6 h, and 24 h for the HaCaT cell line).

### 2.4. Irradiation of Cells

#### 2.4.1. Proton Irradiation

Irradiations with nominal 10 MeV protons were carried out at the PTB ion accelerator facility [25]. A compact cyclotron was used for the production of a proton beam with an initial energy of 14.85 MeV ± 0.08 MeV. To deliver a homogenous proton fluence over the entire cell layer, a part of the primary proton beam was scattered by a 28 µm-thick tantalum foil in the center of a scattering chamber. The cells were positioned at an angle of 90° and at a distance of 183 mm at the bottom in a 48-well plate and at 185 mm at the top of the chamber (Appendix A) in a special dish on 25 µm-thick biofoil (Invitro Systems, Göttingen, Germany). Some well plates contained control samples in an unirradiated well and further control samples were left in the biology laboratory (additional information can be found in Appendix A).

The protons underwent energy losses while passing the scattering foil, the vacuum seal (5 µm and 7 µm molybdenum foil), an air gap of 31 mm and 1 mm of medium at the bottom, or the 25 µm biofoil at the top. A 500 µm aluminum absorber was added on the top in order to provide the same proton energy at both positions after the different energy losses. The final proton energy of the protons at the cell layers was calculated with the software SRIM-2013 [26] at 9.7 MeV ± 0.4 MeV with a calculated LET of 4.8 ± 0.2 keV/µm at bottom and 9.6 MeV ± 1 MeV with a calculated LET of 4.9 ± 0.4 keV/µm at top position (this difference results from the different path of the ions through the scatter foil and a different effective thickness and energy loss). The calculations were based on the nominal thickness of the foils, the measured thickness of the aluminum absorber, and the volume of the medium in the well. The dose applied to the cells was calculated from the LET value and the proton fluence, which was determined from control measurements of scattered protons at both positions and simultaneously monitored with a particle detector at 135 degrees. The integrated proton beam charge, collected at the Faraday cup provided a further monitor. Typical beam currents were 0.3 µA and the typical dose rate was around 30 mGy/min. The uncertainty in the dose values is 7.5% for irradiations at the bottom position and 10.3% at the top. Some well plates contained control samples in an unirradiated well and further control samples were left in the biology laboratory. The interaction of the primary proton beam with the material of slits, the aperture and in the Faraday cup produced a neutron and gamma background. This was measured with a tissue-equivalent ionization chamber (Exradin, Model T2) close to the bottom position of cells at about 14 mGy per Gy of proton dose to the cells.

#### 2.4.2. Gamma Irradiation

Cells were irradiated using a gamma radiation source of Cobalt-60 (^60^Co) located at the National Centre for Scientific Research “Demokritos” in Athens, Greece (Appendix A). The Co-60 source delivers a dose rate of 0.13 Gy/min, ensuring consistent and precise radiation exposure for the samples. Prior to irradiation, the dose rate was verified using a calibrated dosimeter to confirm the accuracy of the applied dose. Cell samples were irradiated under controlled conditions to ensure uniform exposure and reproducibility of results. Additional exposures of normal human skin fibroblast cells (VH10) to gamma radiation at a dose of 0.5 Gy (dose rate: 0.34 Gy/min) were performed using a cesium-137 (^137^Cs) source (GammaCell-1000) at Stockholm University.

#### 2.4.3. UVB Irradiation

A UVB spot source was used to simulate the impact of UVB radiation from sunlight, both independently and in combination with proton radiation, on various normal cell lines (Appendix A). This unit employs a 200 W xenon high-pressure Hamamatsu LC8 lamp. It is equipped with an optical fiber light guide, a filter package to restrict the spectral range mainly to the UVB spectral range, and a light shaping diffuser to produce a uniform irradiation field on the sample box where a typical UV irradiance of 6.6 W/m^2^ could be reached, primarily within the 280 nm to 315 nm wavelength range. The irradiances of the UVB spot were monitored using a UV radiometer situated at the base of the sample holder. The sample holder inside the irradiation chamber allowed for the insertion of the sample box in different orientations and is designed to accommodate up to four slots for exposure. The UVB spot source was assembled, characterized, and repeatedly calibrated at the working group 4.11 “Spectroradiometry” at the PTB. The UVB radiation dose was adjustable via the LC8 Controller software 1.1, enabling the input of specific irradiation durations to regulate the lamp’s output. The necessary exposure time *t* (s) for a given dose $H_{UVB}$ (in J/m^2^) was determined by the following formula:$$t=\frac{H_{UVB}}{E_{UVB}}$$
where $E_{UVB}$ is the actual measured UVB irradiance in (W/m^2^) with the monitor radiometer. Additionally, a UVB radiation source was used for the experiments, specifically the UV Bench Lamp Model XX-15M, with a power output of 15 watts and a peak emission at 302 nm (Appendix A). The lamp was calibrated prior to each experiment to ensure accurate and consistent UVB exposure. Samples were positioned at a fixed distance from the lamp to achieve uniform irradiation. The exposure times and distances were carefully controlled to deliver the desired UVB dose, and all procedures were conducted under standardized laboratory conditions to ensure reproducibility. In some cases, UVB irradiation was performed using a corona mini dose UV240T lamp (230 V, 50 Hz, and 70 W) with an irradiance of 1.4 W/m^2^ for 66 s, corresponding to a fluence of 100 mJ/cm^2^ (additional information can be found in Appendix A).

### 2.5. Post-Exposure Investigations

#### 2.5.1. Post-Exposure Cultivation of Cells

Co-exposed cellular samples and the corresponding controls were cultivated in complete culture medium specific for each cell type, and were cultivated at 37 °C in 5% CO_2_ atmosphere for various tests, as will be described below.

#### 2.5.2. Cell Viability

Cellular viability at various time points after the exposure of cells to protons and/or UVB was assessed as the number of metabolically active cells using the reduction in the tetrazolium compound MTS [3-(4,5-dimethylthiazol-2-yl)-5-(3-carboxymethoxyphenyl)-2-(4-sulfophenyl)-2H-tetrazolium] with the CellTiter 96^®^ AQueous One Solution Cell Proliferation Assay (G3582, Promega Corporation, Madison, WI, USA). Briefly, 10^4^ HaCaT cells, 10^4^ Hs27 cells, or 3 × 10^4^ to −5 × 10^4^ CRL 9855 monocytes were plated in triplicate in a 96-well plate with a flat bottom, in 100 µL total volume of complete culture medium. At various time points, 20 µL of detection reagent from the kit were added in each well. Cells were further cultivated at 37 °C in 5% CO_2_ atmosphere until the colorimetric MTS reduction reaction was developed. The incubation time was fixed for each cell type. In parallel, plasma membrane alteration leading to cell death was assessed as lactate dehydrogenase (LDH) release using the CytoTox 96^®^ Non-Radioactive Cytotoxicity Assay (G1780, Promega Corporation, Madison, WI, USA). Cells were cultivated as described above. At various time points, 50 µL cell-free supernatant was harvested from each sample, and mixed with 50 µL substrate solution from the kit in a 96-well plate with a flat bottom. The colorimetric reaction was allowed to develop for 30 min at room temperature in the dark, and thereafter samples were treated with 50 µL stop reaction from the above-mentioned kit. Both colorimetric reactions were measured as optical density (OD) using a Sunrise Tecan ELISA reader (Tecan, Männedorf, Switzerland) at 490 nm. In the case of the MTS reduction reaction, the 620 nm reference wavelength was used for subtraction. The mean OD of the control samples in triplicate was utilized for calculating the effect of irradiation according to the following formula: irradiation effect = (OD irradiated sample)/(mean OD control).

Supra unit effects mean activation, while subunit effects inhibition.

#### 2.5.3. DNA Damage Immunofluorescence Study

The DNA damage inflicted by the exposure of Hs27 and HaCaT cells to protons or gamma rays and/or UVB was assessed using the γH2AΧ assay as previously described by Tremi et al. [27]. Briefly, exposed and control cells in suspension were plated over 22 × 22 mm^2^ coverslips in 35 mm dishes and were harvested at various time points for the γH2AΧ assay. Cells were fixed with 3% paraformaldehyde (F8775, Sigma-Aldrich, Darmstadt, Germany) and 2% sucrose (A2211, Applichem GmbH, Darmstadt, Germany) in PBS (18912-014, Gibco, Grand Island, NY, USA) for 15 min at RT, and then washed once with PBS. Cells were thereafter permeabilized for 10 min with 0.5% Triton X-100 (X100, Merck, Saint Louis, MO, USA) in 100 Mm Tris-HCL pH 7.4 (A4263, Applichem GmbH, Darmstadt, Germany) and 50 mM EDTA pH 8 (A4982, AppliChem GmbH, Darmstadt, Germany) in distilled water, washed again two times with PBS, and blocked overnight at 7 °C with 0.5% bovine albumin (BSA) (A7906, Sigma-Aldrich, Darmstadt, Germany) and 0.2% gelatin from cold water fish (G7041, Sigma-Aldrich, Darmstadt, Germany) in PBS. Then, samples were stained with an antibody against histone H2AX (p S139) (NB100-384, Novus Biologicals, Abingdon, UK) in 0.5% BSA and 0.2% gelatin in PBS. After a 90 min incubation at RT, cells were washed three times with PBS and were thereafter treated for 90 min at RT with goat anti-rabbit IgG H&L cross-absorbed secondary antibody labeled with Rhodamine Red-X (R-6394, Thermo Fisher Scientific, Waltham, MA, USA) in 0.5% BSA and 0.2% gelatin in PBS. After immunostaining, cells were washed three times with PBS and were stained with ProLong Gold Antifade Reagent with 4′6-diamidino-2-phenylindole (DAPI) (8961, Cell Signaling Technology Inc., Danvers, MA, USA). To access the percentage of γH2AΧ foci corresponding to DSBs, double immunofluorescence (IF) was performed by following the affirmation protocol with a difference in the staining procedure. Samples were double stained with an antibody against histone H2AX (p S139) (80312T, Cell Signaling Technology, Danvers, MA, USA) and an antibody against 53PB1 (88439, Cell Signaling Technology, Danvers, MA, USA) in 0.5% BSA and 0.2% gelatin in PBS. After a 90 min incubation at RT, cells were washed three times with PBS and were after that treated for 90 min at RT with anti-mouse IgG (H+L), F(ab’)2 fragment (Alexa Fluor^®^ 488 Conjugate) (4408S, Cell Signaling Technology, Danvers, MA, USA) and anti-rabbit IgG (H+L), F(ab’) 2 fragment (Alexa Fluor^®^ 594 Conjugate) (8889S, Cell Signaling Technology, 3 Trask Lane, Danvers, MA, USA) in 0.5% BSA and 0.2% gelatin in PBS.

At least two independent experiments were conducted for each experimental condition in all cell lines. Microscope slides were analyzed using a Zeiss Axioskop-2 fluorescence microscope with ISIS 2 software. More than 100 cells/slides were analyzed with the JCountPro and JQuantPlus (Version 1.54p) image processing software (courtesy of Dr. Pavel Lobachevsky group, Peter McCallum Institute, Australia) [28]. Detailed protocols for CRL-9855 and VH10 cell lines are detailed described in the Appendix A).

#### 2.5.4. Transmission Electron Microscopy (TEM)—γH2AX Immunogold Labeling

At 24 h after exposure of cells to gamma rays and UVB, cells were fixed in 3% paraformaldehyde and 0.5% glutaraldehyde in 0.1 M sodium phosphate buffer (PB) for 30 min at RT and then were embedded in 4% gelatin aqueous solution. Specimens (cell-gelatin fragments) were embedded in Lowicryl Hm^2^0 acrylic resin (Polysciences, Warrington, PA 18976, USA) by applying the progressive lowering of temperature (PLT) method [29] in order to better preserve the antigenicity for DNA damage detection. Ultrathin sections (~80 nm) were cut using a Diatome diamond knife (Diatome Ltd., CH-2560 Nidau, Switzerland), and were mounted on formvar-coated Ni grids (200 mesh) and processed for immunogold labeling. Immunocytochemistry was performed using Terasaki-well plates (HLA microtest plates) with lids to ensure a clean dust-free incubation environment and proper humidity. After blocking, sections were incubated overnight at 4 °C with γH2AX primary mouse monoclonal antibody, diluted at 1:100 (80312T, Cell Signaling Technology, Danvers, MA, USA) for single immunolocalization. Immunogold labeling was applied with a 15 nm goat anti-mouse gold-conjugated secondary antibody (810.022, AURION, Wageningen, The Netherlands), diluted at 1:60 for 1 h at room temperature (RT) to give a sufficient specific signal. Afterwards, all sections were stained with uranyl acetate and lead citrate and were observed and photographed using a Jeol JEM-2100Plus Transmission Electron Microscope operated at 80 keV and equipped with a LaB6 filament and a Gatan OneView Camera. Immunogold particles were counted in each condition from 120 electron micrographs at 20,000× original magnification using ImageJ 1.54i 03. A detailed description of the TEM processing and TEM quantification analysis for DNA damage detection can be found in Tremi et al. [27,30].

#### 2.5.5. Chromosomal Aberrations Assays

Lymphocyte culturing was performed in well plates and irradiated as described previously without transfer into the usually used cell culture flasks to prevent cell loss of the just recently irradiated cells. After repair time, 13 µL PHA (Phyothemagglutinin, #10576-015, Gibco-BRL, Dreieich, Germany) was added for lymphocyte stimulation, and incubated for 48 h (37 °C, 5% CO_2_, and 95% humidity). Colcemide (final concentration 0.15 µg mL^−1^; #15212012, Gibco BRL, Germany) was used to block lymphocytes at the metaphase (MP) stage for the last 24 h. Cultures were transferred in 1.5 mL reaction tubes and were centrifuged (200× *g*, 8 min). Hypotonic treatment was performed for 10 min at RT using 1 mL prewarmed (37 °C) 0.075 M potassium chloride (#10575090, Gibco-BRL, Germany). After centrifugation (200× *g*, 5 min), cells were fixed twice with freshly prepared ice-cold fixative (3:1 methanol: acetic acid; 1 mL fixative was added and the solution was pipetted 3 times up and down). Cell suspensions were stored at −20 °C at least overnight. After centrifugation (200× *g*, 8 min) slides were prepared at RT and humidity of about 35% by dispensing 40 µL of a suitable concentrated cell suspension onto clean wet slides and dried for at least 2 h at RT. Dried slides were immersed in Giemsa stain (10% in GURR buffer, pH 6.8; #10582013, Gibco/BRL, Germany) for 3 min. After washing in three changes of A.d., the slides were air-dried for 2 h at RT. MP images were collected using the Metafer4 platform (MetaSystems, Altlussheim, Germany) coupled with a fully motorized Axio Imager Z.1 (Carl Zeiss, Oberkochen, Germany), and a PC controlled microscope for fully automated MP finding, image acquisition, and storing. For semi-automated DCA, scanned slides were run applying the DCScore software module from MetaSystems (version 3.8.101) (Altlussheim, Germany). Chromosomes, indicated as “dicentric” by the DCScore, were evaluated (confirmed or rejected) by a cytogeneticist. MPs were not checked to be complete with 46 centromeres and observed false negative dicentrics were ignored.

For each experiment to examine combined radiation exposures, replicate cell cultures have been set up, resulting in dicentric quantification and determination of standard deviations based on four cell cultures per irradiation condition control cells. At least two slides per culture have been analyzed. Dicentrics induced by 0.5 Gy protons (plus neutron–gamma background radiation), by the neutron–gamma background (plate controls in the irradiation hall), and by 0.5 Gy protons (plus neutron–gamma background) combined with UVB exposure have been quantified. Chi-squared test with one degree of freedom was used to test for significance, comparing the observed dicentric frequency of an examined sample with the PTB laboratory control as well as with the overall laboratory’s reference value for dicentrics in unexposed cells (>10 donors, 14 Cdic in 29265 cells; 0.0005 Cdic/cell) using DoseEstimate software V5.2 [31]. Regarding the HS27 cells, following the 19 h incubation of the samples, colcemid (15212012, Gibco, Grand Island, NY, USA) was added to the cell cultures at a final concentration of 0.1 µg/mL for 5 h to accumulate metaphases for analysis. The cells were then collected through trypsinization and centrifugation, treated in 75 mM KCl for 10 min, and fixed in methanol:glacial acetic acid (3:1 *v*/*v*). Standard procedures [22] were followed for chromosome preparation and Giemsa staining, and chromosomal damage was visualized and quantified as dicentrics in metaphase cells were evaluated. Two experiments were conducted, and approximately 150 cells were scored for chromosome damage for each experimental point based on standard criteria.

#### 2.5.6. Gene Expression

Irradiated and non-irradiated controls of HaCaT and Hs27 cells (0.5 × 10^6^ cells/sample) were cultivated for 48 h in 6-well plates in a total complete medium volume of 2.5 mL, while CRL 9855 monocytes (0.9 × 10^6^ cells/sample) were cultivated for 24 h in 24-well plates in 2 mL complete medium. At the end of the cultivation time, the supernatant of the HaCaT and Hs27 cultures was discarded, the adherent cells, presumed to be viable cells, were washed with Dulbecco’s Phosphate Buffered Saline (DPBS, D8537, Sigma-Aldrich, Saint Louis, MO, USA) and were finally suspended in 1 mL RiboZol™ RNA Extraction Reagent (N580 Avantor, VWR Funding, Radnor, PA, USA). Non-adherent CRL 9855 monocytes were washed by centrifugation in DPBS, and were finally suspended in RiboZol™ RNA Extraction Reagent. All samples in RiboZol were stored frozen at around −18 °C at PTB for 7 days at most, being thereafter transported in dry ice to Romania, where samples were stored at −80 °C until use. Gene expression was evaluated by qRT-PCR, as previously described [32]. Total concentration of RNA in RiboZol samples was measured using the Nanodrop 2000 equipment (NanoDrop Technologies, Wilmington, DE, USA), the registered values of the 260/280 nm and 260/230 nm ratios being above 1.8. The RT^2^ First Strand Kit (Qiagen, Hilden, Germany) was used for cDNA synthesis. The expression of 84 genes involved in stress and toxicity was evaluated using the RT^2^ Profiler™ PCR Array Human Stress and Toxicity Pathway Finder (PAHS-003Z, Qiagen, Hilden, Germany) (Appendix A) and the SYBR Green chemistry on the ABI7500 Fast PCR System (Thermo Fisher Scientific, Waltham, MA, USA) (see also Appendix A). The expression level of each gene was normalized to the geometric mean value of 5 housekeeping genes (ACTB, B2M, GAPDH, HPRT1, and RPLP0). In some cases, relevant housekeeping genes were selected using the RefFinder algorithm [33] out of the five candidate reference genes. Gene expression data were analyzed with the RT^2^ Profiler PCR Array software package 2024 (Qiagen, Hilden, Germany). Gene expression levels were calculated as 2^−ΔCt^ values (Ct = threshold cycle, ΔCt = threshold cycle for the gene–mean threshold cycle for housekeeping genes). Fold change (FC) in gene expression was calculated as the 2^−ΔCt^ values in irradiated samples divided by the 2^−ΔCt^ values in controls. FC values > 1 designate gene over-expression, while FC values < 1 gene under-expression.

## 3. Results

### 3.1. Cellular Viability

#### 3.1.1. Exposure to Protons or/and UVB

The impact of the co-exposure of human Hs27 fibroblasts, HaCaT keratinocytes, and CRL 9855 monocytes to protons and UVB was analyzed in terms of cellular viability in four independent experiments in which cells were exposed first to 0.5 Gy protons and then, within approx. 20 min, to 50 J/m^2^ or 100 J/m^2^ UVB (Figure 2). The responses of cells to single radiation fields (protons or UVB) are described in the Appendix A (Hs27 fibroblasts), Appendix A (HaCaT keratinocytes), and Appendix A (CRL 9855 monocytes).

##### Fibroblasts

Proton exposure slightly decreased at 48 h the MTS reduction by Hs27 fibroblasts, as compared both with CTRL and IrrCTRL (*p* < 0.05) (Figure 2a), while no major effects were detected earlier, at 24 h post-exposure. The observed decrease in MTS reduction was not accompanied by a significant increase in LDH release (Figure 2b), indicating that the decrease in the number of metabolically active Hs27 fibroblasts was not due to significant cell death characterized by alteration of the plasma membrane. While UVB 50 J/m^2^ decreased MTS reduction by Hs27 fibroblasts only at later time points (48 h, *p* < 0.05), higher UVB fluences of 100 J/m^2^ affected MTS reduction both at 24 h and 48 h as compared to CTRL (*p* < 0.01) (Figure 2a). The investigated UVB fluences exerted different effects on Hs27 fibroblasts at 24 h (*p* < 0.01) and 48 h (*p* < 0.001) post-exposure. The LDH released by Hs27 fibroblasts exposed to UVB did not change significantly compared to CTRL (Appendix A), suggesting that the reduction in the number of metabolically active cells (Figure 2a) was not due to significant cell death with alterations in the plasma membrane. The MTS reduction by Hs27 fibroblasts co-exposed to protons (0.5 Gy) and UVB (50 J/m^2^) was significantly lower as compared to the effects of protons or UVB alone (Figure 2a). Thus, the co-exposure of cells induced a slight decrease in MTS reduction at 24 h as compared to protons alone (*p* < 0.05), as well as compared to UVB alone (*p* < 0.05). This decrease was more pronounced at 48 h after co-exposure, both as compared to protons alone (*p* < 0.001) and UVB alone (*p* < 0.05). Altogether, results indicate a persistent synergistic effect of protons and UVB in the co-exposure scenario. When Hs27 cells were co-exposed to 0.5 Gy protons and a higher UVB fluence of 100 J/m^2^, a marked decrease in MTS reduction was registered at 24 h and 48 h after irradiation both against protons alone (*p* < 0.05) and UVB alone (*p* < 0.05) (Figure 2a), indicating again a marked synergistic effect. Meanwhile, no significant changes were registered at the level of LDH release (Figure 2b), indicating again that the decrease in the number of metabolically active cells was not mainly due to cell death with alterations in the plasma membrane integrity (see also Appendix A).

##### Keratinocytes

The impact of the exposure of human HaCaT keratinocytes to 0.5 Gy protons first, and then, within approx. 20 min, to 25 J/m^2^ or 50 J/m^2^ UVB was analyzed in three independent experiments. Protons alone did not alter at 24 h or 48 h MTS reduction as compared either with CTRL or IrrCTRL, while UVB 25 J/m^2^ or 50 J/ m^2^ decreased MTS reduction at 48 h in a fluence–effect relationship (UVB 25 J/m^2^ vs. 50 J/m^2^: *p* < 0.05) (Figure 2b). The co-exposure of HaCaT keratinocytes to 0.5 Gy protons and 25 J/m^2^ UVB did not induce major changes in MTS reduction at 24 h compared to protons or UVB alone (Figure 2b). Instead, it induced a decrease in MTS reduction later, at 48 h post-exposure, as compared to protons alone (*p* < 0.05), but not as compared to UVB alone (Figure 2b), indicating that the main effect was due to UVB. When the UVB fluence of 50 J/m^2^ was used in the co-exposure setting with 0.5 Gy protons (Figure 2b), MTS reduction by HaCaT keratinocytes was significantly lower in co-exposed samples compared to protons alone both at 24 h (*p* < 0.05) and 48 h (*p* < 0.01), but was comparable to the data registered in samples exposed to UVB alone (Figure 2b), indicating that the effect was mainly due to the UVB exposure.

LDH release by HaCaT cells co-exposed to 0.5 Gy protons and 25 J/m^2^ or 50 J/m^2^ UVB was increased (mean effect > 1) in both UVB irradiation settings (Appendix A), but differences were not statistically significant between the LDH released when cells were co-exposed to protons and UVB or were challenged with single radiations. Nevertheless, the cellular systems at 48 h post-exposure were unstable regarding LDH release in independent experiments both in the case of Hs27 fibroblasts (Figure 2b) and HaCaT keratinocytes (Figure 2d), limiting the significance of the results (see also Appendix A).

##### Monocytes

The co-exposure of CRL 9855 monocytes to 0.5 Gy protons, and approx. 20 min thereafter, to 50 J/m^2^ UVB induced at 24 h post-exposure marked a decrease in MTS reduction that was due to the UVB exposure (Figure 2c). While UVB alone decreased to approx. 40% the MTS reduction, protons alone did not have a notable effect in the co-exposure experiments (Figure 2c). The UVB-induced decrease in the number of metabolically active cells was not accompanied by an increase in LDH release (Figure 2c), indicating that it was not most probably due to cell death with alteration in the plasma membrane integrity.

Altogether, 0.5 Gy of protons alone did not induce significant alteration in cellular viability, while 50 J/m^2^ UVB had more pronounced effects, especially in monocytes. Synergistic effects were registered in various co-exposure settings applied to human skin cells, while in monocytes that are more sensitive to UVB, the response to UVB was dominant (see also Appendix A).

#### 3.1.2. Exposure to Gamma Rays or/and UVB

The impact of UVB and gamma rays as single or combined radiations on human normal skin cells (Hs27 fibroblasts and HaCaT keratinocytes) was investigated for comparison with the proton-UVB irradiation setting, described above. Cell viability was assessed using the MTS reduction assay that provides information on the number of metabolically active cells in culture.

##### Fibroblasts

The impact of the co-exposure of Hs27 cells to gamma rays and UVB was analyzed. In two independent experiments, we investigated the viability of Hs27 fibroblasts exposed first to 0.5 Gy gamma rays and then, within approx. 20 min, to 100 J/m^2^ UVB. In this set of experiments, gamma rays did not significantly decrease MTS reduction at both 24 h and 48 h, as compared with the CTRL (Figure 2d). In turn, 100 J/m^2^ UVB affected MTS reduction at 24 h compared to CTRL (*p* < 0.05), while at 48 h, results show no significant difference (Figure 2d). While the exposure of cells to 0.5 Gy gamma rays and thereafter to 100 J/m^2^ UVB did not induce a decrease in MTS reduction at 24 h as compared to gamma rays or UVB 100 J/m^2^ alone, a pronounced decrease was registered at 48 h after co-exposure, both as compared to gamma rays alone (*p* < 0.01) and UVB alone (*p* < 0.01) (Figure 2d). These findings agree with the results obtained in the proton and UVB co-exposure setting.

##### Keratinocytes

The impact of the exposure of HaCaT cells to 0.5 Gy gamma rays first, and then within approx. 20 min, to 50 J/m^2^ UVB was analyzed in two independent experiments. In this set of co-exposure experiments, gamma rays alone or 50 J/m^2^ UVB alone did not alter at 24 h or 48 h of the MTS reduction as compared with CTRL, and neither did the co-exposure of cells to 0.5 Gy gamma rays and 50 J/m^2^ UVB (Figure 2e), indicating that keratinocytes are resistant to the action of this type of combined exposure.

### 3.2. DNA Damage

#### 3.2.1. Exposure to Protons and/or UVB

DNA damage and repair were assessed using γH2AΧ IF in Hs27 fibroblasts and HaCaT keratinocytes exposed to 0.5 Gy protons or monocytes exposed to 0.25 Gy, and UVB radiation (50 J/m^2^ or 100 J/m^2^ for fibroblasts, 25 J/m^2^ or 50 J/m^2^ for keratinocytes, and 100 J/m^2^), either as single or combined radiations. Two experimental settings regarding the succession of the co-exposure of cells to protons and UVB were investigated: in one setting, cells were exposed to protons first, and then, within approx. 20 min, to UVB; in another setting, cells were exposed first to UVB, and then to protons. Both the number of γH2AΧ foci per nucleus and pan-nuclear-stained cells were analyzed. Irradiated cells and controls were analyzed at 1 h, 4 h, and 24 h post-exposure of fibroblasts, at 1 h, 6 h, and 24 h post-exposure of keratinocytes, and at 1 h and 24 h post-exposure of monocytes.

##### Fibroblasts

Irradiation of Hs27 fibroblasts with 0.5 Gy protons induced an increased number of γH2AΧ foci per nucleus at 1 h post-exposure that declined in time, as observed at 4 h and 24 h (Figure 3a). Meanwhile, UVB (50 J/m^2^ and 100 J/m^2^) increased persistently the number of γH2AΧ foci per nucleus at 4 h and 24 h post-exposure (Figure 3a). When cells were exposed first to protons and then to UVB, a synergistic effect of these radiations was observed (Figure 3a). Thus, cells exposed to the combination of 0.5 Gy protons and 100 J/m^2^ UVB displayed markedly higher levels of γH2AX foci per nucleus compared to cells exposed to protons or UVB alone, indicating a potential detrimental synergistic effect of radiations in Hs27 fibroblasts at all time points (*p* < 0.05 or *p* < 0.01). Samples exposed to the combination of 0.5 Gy protons and lower UVB fluences of 50 J/m^2^ displayed higher levels of γH2AX foci per nucleus compared to cells exposed to protons or UVB alone (*p* < 0.05 or *p* < 0.01), except at the 24 h time point, where the difference between UVB-exposed and co-exposed samples was not statistically significantly different (*p* > 0.05).

Pan-nuclear-stained Hs27 fibroblasts relative to γH2AX were counted, and the percentage was analyzed (Figure 3b), considering that pan-nuclear γH2AX staining mediated by ATM or DNA-PK phosphorylation shows excessive stress due to highly repair-resistant complex damage, leading usually to apoptotic cell death [34]. Cells exposed to 0.5 Gy protons alone showed minimal pan-nuclear γH2AX staining across all time points. In contrast, UVB exposure (50 J/m^2^ and 100 J/m^2^) induced a notable increase in pan-nuclear staining, particularly at 1 h and 4 h, with a decline later, at 24 h. Co-exposure of cells to 0.5 Gy protons, and either to 50 J/m^2^ or 100 J/m^2^ UVB, significantly amplified the increase in the percentage of pan-nuclear-stained cells in comparison with cells exposed to single radiation, especially at 1 h (42.28% and 49.12% with 50 J/m^2^ or 100 J/m^2^ UVB, respectively) and 4 h (50.91% and 64.71% with 50 J/m^2^ or 100 J/m^2^ UVB, respectively). The highest percentage of pan-nuclear staining was observed in the samples co-exposed to 0.5 Gy and 100 J/m^2^ at the 4 h time point. Results indicate a synergistic enhancement of the DNA damage response (DDR) in co-exposed Hs27 fibroblasts, and that the combination of proton irradiation and UVB exposure elicited a marked increase in pan-nuclear γH2AX staining, reflecting broadly enhanced DNA damage and repair signaling in fibroblasts. Representative fluorescence images are presented in Figure 3f.

In the second irradiation setting, cells were first exposed to UVB and then to protons and analyzed at 4 h and 24 h post-exposure. The number of γH2AX foci per nucleus was slightly elevated after 4 h, but significantly decreased later, at the 24 h time point in Hs27 fibroblasts exposed to 0.5 Gy protons applied as single radiation (Appendix A). Results indicate a reduction in DNA damage over time, as cells attempt to repair proton-induced DSBs. Cells exposed to 100 J/m^2^ UVB showed a similar trend, with an increase in γH2AX foci at 4 h, followed by a reduction at 24 h (Appendix A), suggesting a typical DNA damage response to UVB exposure [35]. In the combined exposure setting of 100 J/m^2^ and 0.5 Gy protons, the number of γH2AX foci per nucleus was markedly higher compared to either treatment applied alone (*p* < 0.05 or *p* < 0.01). The average number of foci reached ~5.0 foci/nucleus at 4 h, and decreased to ~3.5 foci per nucleus at 24 h post-exposure (Appendix A). The significant increase in γH2AX foci suggests a synergistic effect when protons and UVB are combined, leading to enhanced DNA damage.

A very low percentage of pan-nuclear-stained cells with γH2AX staining was detected in Hs27 fibroblasts exposed only to 0.5 Gy protons at 4 h and 24 h post-irradiation (Appendix A), suggesting minimal occurrence of global DNA damage across the nucleus. Meanwhile, 100 J/m^2^ UVB induced a considerably higher percentage of pan-nuclear γH2AX staining (~50% cells stained at 4 h), with a slight decrease by 24 h (Appendix A), indicating that UVB alone causes substantial pan-nuclear DNA damage that decreases over time due to active repair processes. The highest percentage of pan-nuclear staining was observed in the samples co-exposed to 0.5 Gy and 100 J/m^2^ at the 24 h time point.

Comparison between the two co-exposure scenarios revealed no statistically significant differences at any of the investigated time points (4 h and 24 h) (*p* > 0.05). This indicates that, despite the observed distinct trends in DNA damage and repair, the differences in γH2AX foci between the two exposure sequences were not statistically significant and could be attributed currently to experimental variability mostly. Therefore, the overall response to combined radiation exposure, whether protons precede UVB or vice versa, seems to result in comparable levels of DNA damage and repair kinetics, although there might be a stronger effect in the second scenario (additional information can be found in Appendix A).

##### Keratinocytes

Irradiation of HaCaT cells with 0.5 Gy protons caused a substantial increase in γH2AX foci, peaking at 1 h post-exposure and gradually decreasing over 6 h and 24 h (Figure 3c). This indicates an initial strong induction of DNA DSBs by proton irradiation, with a subsequent repair process reducing the foci over time. When HaCaT cells were exposed to UVB (25 J/m^2^ and 50 J/m^2^), a moderate increase in γH2AX foci was observed, with the response being more pronounced at the higher UVB dose. Meanwhile, UVB persistently increased the number of γH2AΧ foci per nucleus later, at 6 and 24 h post-exposure (Figure 3c). Notably, when cells were exposed first to protons, and within approx. 20 min, to UVB, a synergistic effect of these radiations was observed (Figure 3c). The number of γH2AX foci significantly increased compared to either treatment alone at 6 h post-exposure (*p* < 0.05 or *p* < 0.01), particularly when cells were co-exposed to protons and the higher UVB dose. This synergistic effect was still evident at 24 h, as the number of γH2AX foci was significantly higher in cells co-exposed to 0.5 Gy protons and 50 J/m^2^ UVB compared to either treatment alone (*p* < 0.05 or *p* < 0.01). Meanwhile, the co-exposure to protons and the lower UVB fluence (25 J/m^2^) resulted in a significantly higher number of γH2AΧ foci only compared to the number of γH2AX foci caused by proton exposure or UVB at the 6 h time point (*p* < 0.05 or *p* < 0.01), while at 24 h, there were no statistically significant difference between co-exposed and single irradiated samples.

The impact of combined proton and UVB irradiation on HaCaT cells was further evaluated by measuring the percentage of pan-nuclear-stained cells using the γH2AX antibody as a marker for DNA damage (Figure 3d). The percentage of pan-nuclear-stained cells that were previously exposed to 0.5 Gy protons remained relatively low across all time points, suggesting a limited pan-nuclear DNA damage response to proton irradiation alone. However, exposure of cells to UVB alone (25 J/m^2^ or 50 J/m^2^) led to a substantial increase in the percentage of pan-nuclear-stained cells, with the effect being more pronounced at 50 J/m^2^ UVB, as expected. This increase was particularly notable at 1 h post-exposure, and remained elevated at 6 h, though it began to decline by 24 h, indicating a peak in the DNA damage response followed by partial repair. Combined exposure results emphasize a marked synergistic effect. The percentage of pan-nuclear-stained cells in the co-exposure setting was higher than in either treatment alone, particularly at 1 h and 6 h post-exposure. This synergistic response was especially obvious in cells co-exposed to 0.5 Gy and 100 J/m^2^ at the 6 h time point, where the pan-nuclear staining remained elevated even at 24 h, highlighting a sustained DNA damage response. These findings demonstrate that while UVB irradiation alone can induce considerable pan-nuclear DNA damage in HaCaT cells, the addition of proton irradiation significantly amplified this effect, leading to a more robust and persistent response. Representative fluorescence images are presented in Figure 3g.

In the second irradiation setting, cells that were first exposed to UVB and then to protons were analyzed at 6 h and 24 h post-exposure. A noticeable increase in γH2AX foci was observed at 6 h post-exposure, indicating a significant induction of DSBs, but the number of foci decreased after 24 h, suggesting active DNA repair mechanisms reducing the initial damage (Appendix A). Exposure of cells to 50 J/m^2^ of UVB alone produced a moderate increase in γH2AX foci, with a similar trend of peak foci at 6 h, followed by a reduction at 24 h (Appendix A). Importantly, the combined exposure to 0.5 Gy protons and 50 J/m^2^ UVB led to a synergistic effect, with the highest number of γH2AX foci observed at 6 h (Appendix A). This exposure setting resulted in significantly more DNA damage than either treatment at the 6 h time point, as evidenced by the elevated γH2AX foci. Although the foci count decreased at 24 h (Appendix A), their number remained higher than the levels observed in individual exposures (*p* < 0.05 or *p* < 0.01).

The percentage of pan-nuclear-stained cells remained relatively low, with a slight increase observed at 24 h, suggesting that proton irradiation induced limited pan-nuclear DNA damage (Appendix A). Meanwhile, exposure to 50 J/m^2^ UVB led to a more pronounced increase in the percentage of pan-nuclear-stained cells, particularly at 6 h when the damage response was apparently at its peak. There was a noticeable reduction in pan-nuclear staining at 24 h, indicating that keratinocytes were able to initiate DNA repair processes to mitigate the UVB-induced damage (Appendix A). However, the most significant finding was observed when cells were exposed to 50 J/m^2^ UVB followed by 0.5 Gy protons. This combined exposure resulted in a dramatic increase in the percentage of pan-nuclear-stained cells at 6 h, much higher than the levels induced by either treatment alone. Even at 24 h, the levels remained substantially higher compared to the individual exposures, although there was a reduction in pan-nuclear staining (Appendix A). These results highlight the synergistic effects of combined UVB and proton irradiation that lead to significantly enhanced and sustained complex DNA damage compared to individual treatments. The marked increase in pan-nuclear staining following combined exposure suggests that keratinocytes experienced more extensive and persistent DNA damage, likely due to the combined effects of the two types of radiation.

The two co-exposure scenarios were compared, and the results show statistically significant differences at the two time points (4 h and 24 h) (*p* < 0.05 or *p* < 0.01). When cells were first exposed to protons followed by UVB, there was a strong and immediate increase in DNA damage, with the damage being sustained over time, particularly in the case of the higher UVB dose (50 J/m^2^). This suggests that proton irradiation sensitizes HaCaT keratinocytes, making them more vulnerable to subsequent UVB-induced damage. Conversely, when UVB was followed by proton exposure, the initial DNA damage response to UVB was less pronounced, but the subsequent proton exposure amplified the damage, leading to a significant increase in γH2AX foci. These results indicate that, while UVB alone can induce DNA damage, the addition of protons exacerbates the effect, resulting in more persistent damage (additional information can be found in Appendix A).

##### Monocytes

Irradiation of CR 9855 cells with 0.25 Gy protons caused a substantial increase in γH2AX foci, peaking at 1 h post-exposure, and gradually decreasing over 24 h (Figure 3e). This indicates an initial strong induction of DSBs by proton irradiation, with a subsequent repair process reducing the foci over time. When monocytes were exposed to 100 J/m^2^ UVB, a moderate increase in γH2AX foci was observed. Notably, when cells were exposed first to protons, and within approx. 20 min, to UVB, a synergistic effect of these radiations was observed (Figure 3e). The number of γH2AX foci was increased compared to either treatment alone, and this synergistic effect persisted over 24 h, with a statistically significant difference both as compared to protons and UVB alone (*p* < 0.05 or *p* < 0.01).

#### 3.2.2. Exposure to Gamma Rays or/and UVB

DNA damage and repair were assessed using γH2AX IF in human Hs27 and VH10 fibroblasts, and in HaCaT keratinocytes exposed to a dose of 0.5 Gy gamma rays, and within 20 min, to 100 J/m^2^ UVB for fibroblasts, and 50 J/m^2^ UVB for keratinocytes, applied either individually or in combination (Figure 4e,f). The number of γH2AX foci per nucleus in the irradiated cells and controls was analyzed at 1 h, 4 h, and 24 h post-exposure for fibroblasts, and at 1 h, 6 h, and 24 h post-exposure for keratinocytes.

##### Fibroblasts

Gamma irradiation of Hs27 fibroblasts induced a significant synergistic enhancement in DNA damage when cells were exposed to a combination of gamma rays and UVB radiation compared to single exposures. The combined exposure condition of 0.5 Gy protons and 100 J/m^2^ UVB consistently resulted in a higher number of γH2AX foci per nucleus across all time points, with a peak at 1 h post-exposure, indicating an immediate and substantial effect of irradiation on DNA integrity (Figure 4a). While DNA repair processes were obvious, as shown by the decrease in γH2AX foci from 1 h to 24 h, the combined exposure still induced elevated DNA damage levels at 24 h, suggesting that the synergistic effect may impede repair mechanisms or cause more complex damage. The comparison between single and combined exposures revealed that, while both 0.5 Gy gamma radiation and 100 J/m^2^ UVB independently increased DNA damage, the combination resulted in more severe damage, with a statistically significant difference at the 24 h time point compared to single irradiations (*p* < 0.05 or *p* < 0.01).

The percentage of pan-nuclear-stained cells was analyzed (Figure 4b), indicating a pronounced synergistic effect, particularly under combined exposure to gamma rays and UVB radiation (0.5 Gy and 100 J/m^2^). This synergistic response was most evident at the early time points, with a significant increase in the percentage of pan-nuclear-stained cells observed as early as 1 h post-exposure. Notably, the single exposures to 0.5 Gy resulted in low percentages of pan-nuclear-stained cells, while exposure to 100 J/m^2^ caused a similar percentage as the combination of gamma rays and UVB. Over time, there was a trend towards a plateau in the percentage of pan-nuclear-stained cells, indicating that fibroblasts may be experiencing sustained or irreparable damage, leading to prolonged pan-nuclear staining. Representative fluorescence images are presented in Figure 4e.

Transmission electron microscopy (TEM) analysis was performed to investigate the formation of DSBs by applying single immunogold localization of γH2AX in Hs27 fibroblasts co-exposed to 0.5 Gy gamma rays and 100 J/m^2^ UVB in comparison to the non-irradiated control samples. In co-exposed fibroblasts (Figure 5a), the number of γH2AX particles per μm^2^ of nuclear surface area was significantly elevated, with an average of 1.35 particles/nuclear area (μm^2^), compared to the control sample (Figure 5b), which exhibited an average of 0.29 particles/nuclear area (μm^2^) (Figure 5c). Moreover, 48.75% of the analyzed nuclear areas in the co-exposed samples exhibited γH2AX gold particle clusters, while only 3.9% (only one cluster per electron micrograph) of the nuclear areas in the control sample registered with clusters. These findings demonstrate a substantial increase in γH2AX gold particle density per μm^2^ of nuclear surface area in co-exposed fibroblasts, reinforcing the conclusion that combined exposure to gamma rays and UVB induces a synergistic DNA damage response that is markedly higher than the baseline levels observed in non-irradiated cells even after 24 h of repair.

DNA damage signaling mediated by γH2AX and the repair mechanism mediated by the key regulator 53BP1 [36] were analyzed through a co-localization study in Hs27 fibroblasts. Results reveal significant differences in cellular responses to different exposure conditions. In Hs27 fibroblasts exposed to 0.5 Gy gamma rays, high percentages of colocalization were observed, ~79% at the 1 h time point, decreasing to ~62% at 4 h, and further to ~47% at 24 h (Figure 5d). These high levels of colocalized γH2AX and 53BP1 indicate a robust recruitment of DNA damage response proteins to DSBs. In contrast, fibroblasts exposed to 100 J/m^2^ UVB radiation exhibited substantially lower colocalization, almost background levels, consistent with UVB-induced γH2AX forming a pan-nuclear pattern rather than discrete foci. Interestingly, co-exposure to 0.5 Gy gamma rays and 100 J/m^2^ UVB resulted in high percentages of colocalization compared to UVB-exposed samples, but significantly lower at all time points compared to samples exposed to gamma rays (~41% at the 1 h time point, ~37% at the 4 h time point, and ~30% at the 24 h time point) (Figure 5e), indicating that co-exposed cells are less prone to repair DNA damages inflicted by gamma rays.

Gamma irradiation of human VH10 fibroblasts induced as well a significant synergistic enhancement in DNA damage when cells were exposed to a combination of gamma rays and UVB radiation compared to single exposures (Appendix A). The combined exposure condition of 0.5 Gy gamma rays and 100 J/m^2^ UVB consistently resulted in a higher number of γH2AX foci per nucleus across at all time points, with a peak at 1 h post-exposure, indicating an immediate and substantial effect of irradiation on DNA integrity (Appendix A). While DNA repair processes were evident, as shown by the decrease in γH2AX foci from 1 h to 24 h, the combined exposure still induced elevated DNA damage levels at 24 h. This suggests that the synergistic effect may impede repair mechanisms or cause more complex damage. The comparison between single and combined exposures revealed that both 0.5 Gy gamma radiation and 100 J/m^2^ UVB independently increased DNA damage, but the combination resulted in significantly greater damage across all time points, with a statistically significant difference observed for co-exposure versus single exposures (*p* < 0.05 or *p* < 0.01 for all comparisons).

Pan-nuclear detection of γH2AX in VH10 fibroblasts was performed, and the percentage of stained cells was analyzed (Appendix A). Similar to the response observed in Hs27 fibroblasts, the results indicate a pronounced synergistic effect on the induction of pan-nuclear γH2AX staining, particularly under combined exposure to gamma rays and UVB radiation (0.5 Gy + 100 J/m^2^). This synergistic response was evident at all time points, with a significant increase in the percentage of pan-nuclear-stained cells observed as early as 1 h post-exposure. Single exposures to either 0.5 Gy gamma radiation or 100 J/m^2^ UVB alone resulted in relatively low percentages of pan-nuclear-stained cells, further emphasizing the ability of combined exposure to trigger a widespread DNA damage response. Over time, a plateau was observed in the percentage of pan-nuclear-stained cells in the combined exposure group, suggesting sustained or irreparable damage, leading to prolonged pan-nuclear staining. These findings demonstrate that VH10 fibroblasts exhibit similar sensitivity to DNA damage from combined radiation exposures. Fluorescence images of VH10 cells can be found in Appendix A.

##### Keratinocytes

The analysis of γH2AX foci in HaCaT keratinocytes (Figure 4c) revealed distinct cellular responses under different exposure conditions (0.5 Gy gamma radiation, 50 J/m^2^ UVB radiation, and their combination). Following 0.5 Gy gamma exposure, the number of γH2AX foci was highest at 1 h post-exposure, and decreased over time, while following UVB exposure, the number of γH2AΧ foci was low in all time points. The combined exposure of 0.5 Gy gamma radiation and 50 J/m^2^ UVB did not result in statistically higher γH2AX foci compared to a single exposure to gamma rays, with a peak at 1 h. These results do not align with observations in Hs27 cells, meaning that there is no clear indication of a consistent synergistic effect.

The analysis of pan-nuclear γH2AX staining in HaCaT keratinocytes (Figure 4d) revealed a higher sensitivity of keratinocytes to DNA damage as compared to fibroblasts. Exposure to 0.5 Gy gamma radiation alone resulted in consistently low percentages of pan-nuclear-stained cells across all investigated time points, indicating minimal induction of pan-nuclear damage by this radiation dose. In contrast, exposure to 50 J/m^2^ UVB led to a significant increase in the percentage of pan-nuclear-stained cells, with a peak at 1 h post-exposure, followed by a gradual decrease over time, suggesting that UVB exposure induced a stronger DNA damage response compared to gamma radiation. The combined exposure of 0.5 Gy gamma radiation and 50 J/m^2^ UVB showed in this case similar levels of pan-nuclear-stained cells compared to samples exposed to UVB, indicating that the UVB cue dictated cellular responses. Representative fluorescence images are presented in Figure 4f.

### 3.3. Genomic Instability–Chromosomal Aberrations Assay

The γH2AX assay is a sensitive and early indicator of DSBs, providing rapid insights into genotoxic effects. However, while the γH2AX biomarker detects DNA damage, it does not distinguish between transient and persistent lesions or indicate whether the damage leads to chromosomal abnormalities. The chromosomal aberration (CA) analysis complements γH2AX assay by assessing structural and numerical chromosomal changes, which are critical for evaluating long-term genetic stability and potential carcinogenic risk. Performing the CA assay after γH2AX ensures a comprehensive analysis of genotoxic effects, distinguishing between DNA damage response activation and actual chromosomal alterations.

#### 3.3.1. Exposure to Gamma Rays or/and UVB

CA yields observed in cells exposed to the 0.5 Gy gamma rays (0.23 CA/cell) or to 100 J/m^2^ UVB (0.32 CA/cell) were not significantly different (*p* > 0.05) from the background CA yield in unexposed cells (0.2 CA/cell). The CA yields in cells exposed to 0.5 Gy gamma rays followed immediately by 100 J/m^2^ UVB were increased about 1.5-fold (0.65 AC/cell), compared to the chromosomal aberration yield in gamma rays-irradiated cells (Figure 5a). The differences were significant compared to control cells and to single irradiated samples (*p* < 0.05 or *p* < 0.01). Gamma irradiation of Hs27 cells demonstrated a significant synergistic enhancement in DNA damage when cells were exposed to a combination of gamma rays and UVB radiation compared to single exposures. The frequency of chromatid exchanges was analyzed in human normal fibroblasts exposed to 0.5 Gy gamma rays, 100 J/m^2^ UVB, and their combination. The data reveal a stark difference in chromatid exchange frequencies across the experimental conditions. In control samples and samples exposed to 0.5 Gy gamma rays alone, no chromatid exchanges were observed. In contrast, UVB-exposed samples exhibited a low frequency of chromatid exchanges, with 2.27% of cells displaying these aberrations, while co-exposure to 0.5 Gy gamma rays and 100 J/m^2^ UVB significantly increased the frequency of chromatid exchanges to 7.68% (Figure 5b).

#### 3.3.2. Proton or/and UVB Induced Dicentric Frequencies in Lymphocytes

The dicentric chromosome assay (DCA) in lymphocytes is a well-established method for assessing radiation-induced damage at the cytogenetic level, specifically identifying unstable chromosomal aberrations caused by misrepaired double-strand breaks.

UVB exposure of lymphocytes did not induce dicentric formation significantly different to the laboratory’s unexposed control cells from 100 J/m^2^ up to 400 J/m^2^ with the observation that after 400 J/m^2^, only a few cells were still viable and proliferative. The dicentric yields observed in the PTB laboratory controls and in the UVB-exposed cells (0.001 Cdic/cell after 400 J/m^2^ and 0.0006 Cdic/cell after 100 J/m^2^ UVB) were not significantly different (*p* > 0.001) from the background dicentric yield in unexposed cells of the analyzing laboratory (0.0005 Cdic/cell). However, a significant dicentric induction was found in the plate controls of combined exposure experiments (0.0121 CdicNG/cell, *p* < 0.001; 0.0020 CdicNG/cell, *p* < 0.001) as well as in all other proton and proton/UVB or UVB/proton exposed samples (*p* < 0.001) (Appendix A and Figure 6c,d). However, exposure to UVB 400 J/m^2^ followed by 0.5 Gy protons was the most harmful scenario, leading to too few viable cells for a reliable determination of the induced dicentric frequency (Appendix A).

The dicentric yield in cells exposed to 0.5 Gy protons followed immediately by 100 J/m^2^ or 400 J/m^2^ UVB was increased about 2-fold or 1.5-fold, respectively, compared to the dicentric yield in proton-irradiated cells (Figure 6c). When cells were first exposed to 100 J/m^2^ UVB and then to 0.5 Gy protons, a 1.3-fold elevation of the dicentric yield after the combined exposure was observed compared to 0.5 Gy proton irradiation only (Appendix A). As upon irradiation with 400 J/m^2^ UVB first followed by 0.5 Gy protons, the metaphase yield was too low to be reliably analyzed for dicentrics, so no fold induction of dicentric formation has been determined. Accordingly, upon irradiation with 100 J/m^2^ UVB first followed by 0.5 Gy protons, the metaphase yield was also reduced compared to exposure to protons first followed by 100 J/m^2^ UVB (see also Appendix A).

### 3.4. Stress Genes Expression Assay

Fibroblasts

The expression changes of stress genes were quantitatively investigated by qRT-PCR using a pathway-focused array in Hs27 fibroblasts, HaCaT keratinocytes, and CRL 9855 monocytes exposed to 0.5 Gy protons or 50 J/m^2^ UVB as single radiations, or co-exposed first to protons, and within approx. 20 min thereafter, to UVB. Only adherent Hs27 and HaCaT cells, presumably living cells, were investigated, while in the case of non-adherent CRL 9855 monocytes, all cells, both living and dead, were analyzed. Late events registered at 48 h for fibroblasts and keratinocytes, and at 24 h for non-adherent monocytes were investigated as prolonged stress responses. Gene expression results are expressed as “fold change” (FC) relative to the control kept in the biology laboratory (CTRL). Genes with FC values > 1.5 were considered significantly up-regulated, while genes with FC values < 0.7 were considered significantly down-regulated.

When Hs27 fibroblasts were co-exposed to 0.5 Gy protons and 50 J/m^2^ UVB, a broader pattern of gene expression changes was evidenced (Figure 7a) as compared to the patterns found when cells were exposed solely to protons (Appendix A) or to UVB (Appendix A). For instance, while *GADD45G* was down-regulated in all the investigated irradiation settings, only in the co-exposure scenario *GADD45A* was it found to be up-regulated, indicating a prolongation of the GADD45-mediated responses in co-exposed fibroblasts, in addition to *CDKN1A* and *HUS1* as cell cycle blockers. XPC was up-regulated no matter if cells were exposed only to protons (Appendix A) or to UVB (Appendix A) or were subjected to combined radiations (Figure 7a).

Several other stress genes were found up-regulated at 48 h after exposure of Hs27 fibroblasts to protons, UVB, or to combined radiations (Appendix A). These genes are involved in cellular responses to oxidative stress (*GSR* [37]), hypoxia (*ARNT* [38]), and inflammation (*IL1A* and *IL1B* [39]). There was no statistical difference regarding the expression changes of common genes among irradiation settings, indicating that radiation effects were not additive with respect to the stress response mediated by these genes.

Keratinocytes

When HaCaT keratinocytes were exposed to 0.5 Gy protons (Appendix A) or to 50 J/m^2^ UVB (Appendix A), *DDB2* and *XPC* (*XPC* FC: 6.30 ± 8.03) were up-regulated, indicating an active DNA damage response at 48 h post-exposure. Meanwhile, several other genes involved in the DNA damage response were down-regulated in UVB-exposed cells (Appendix A), such as *GADD45A/B* and *CHEK2*, as found also in the case of proton exposure (Appendix A). The *RAD9A* gene involved in cell cycle arrest [40] was found down-regulated specifically in the UVB exposure setting. Moreover, the UVB-specific up-regulation of *CASP1*, which encodes a caspase involved in pyroptotic cell death mediated by inflammatory cytokines [41], was registered. The *TNFRSF10A/B* genes were down-regulated in cells exposed to UVB (Appendix A), as found also in cells irradiated with protons (Appendix A), indicating a lower susceptibility of cells to receptor-mediated cell death at 48 h post exposure.

The pattern of stress genes related to DNA damage and cell death, which were found up-regulated in HaCaT keratinocytes co-exposed to protons and UVB (Figure 7b), recapitulate partly the pattern inflicted by protons (Appendix A) or by UVB (Appendix A) alone. Thus, *ATM* and *DDB2* were up-regulated in co-exposed cells, as they were in proton-irradiated keratinocytes, while *CASP1* was up-regulated in co-exposed cells, as it was in UVB-exposed cells (Appendix A). *ATR* was up-regulated only in co-exposed cells, reinforcing the DNA breaks sensing of *ATM* and *XPC* (XPC FC: 7.96 ± 10.94). In turn, *CHEK2* and *GADD45* genes were down-regulated both in co-exposed, and proton- (Appendix A) or UVB-irradiated cells (Appendix A). Meanwhile, *RAD9A* was down-regulated in co-exposed keratinocytes, as it was also in cells subjected to UVB (Appendix A), but not to protons (Appendix A). The down-regulation of *RIPK1*, involved in necroptosis, and of *ULK1*, involved in autophagy, was detected solely in co-exposed cells, indicating that co-exposed cells were not committed to these types of cell death.

In addition to DNA damage and cell death genes, several other stress genes had modified expression HaCaT keratinocytes in the investigated exposure settings (Appendix A).

An antioxidant response was highlighted (Appendix A) by the over-expression of *FTH1* (iron metabolism), *GSTP1* (glutathionylation), and *PRDX1* (reduction in hydrogen peroxide and alkyl hydroperoxides). Only *FTH1* was selectively up-regulated in the co-exposure setting as compared with the exposure to protons or UVB as single stressors (*p* < 0.05 or *p* < 0.01), indicating increased intracellular iron storage. The expression changes of the other antioxidant genes mentioned above were similar in all the irradiation scenarios. Several genes involved in the hypoxia response (*ADM*, *SERPINE1*, and *SLC2A1*) were down-regulated, while the *ARNT* gene was up-regulated across all the investigated settings (co-exposure: FC = 8.1 ± 9.28; protons: FC = 6.78 ± 5.19; UVB: FC = 12.03 ± 18.28), with no statistical differences between conditions (Appendix A). Results indicate that the hypoxia response had active components, but other associated mechanisms were profoundly down-regulated at 48 h post-exposure of HaCaT keratinocytes. Down-regulations were registered also in the pro-apoptotic *BBC3* gene [42] or *DNAJC3* gene encoding the heat shock protein *HSP40*, part of the unfolded protein response [43] (Appendix A). Down-regulation was registered across all the investigated exposure types, irrespective of whether stressors were applied alone or in combination. The over-expression of *IL1B* [44] and *MCL1* [45] (Appendix A) indicated that inflammation was active in exposed keratinocytes irrespective of the irradiation setting. The concomitant over-expression of CASP1 is indicative that the *IL-1* response might be indeed fully active [46]. In turn, *TNF* [47] was found down-regulated (Appendix A), indicating a limitation in the inflammatory response triggered by radiation at 48 h post-exposure.

Monocytes

Most of the down-regulated stress genes in proton exposure were found down-regulated in cells exposed to 50 J/m^2^ UVB applied as a single stressor, and in cells co-exposed to protons and UVB (Appendix A). There was only one notable exception: the *TNFRSF1* gene [48] involved in necrotic cell death, which was down-regulated only in monocytes co-exposed to protons and UVB, while single stressors did not significantly affect its expression (*p* < 0.05 or *p* < 0.01) (Appendix A). The stress response of human normal CRL 9855 monocytes exposed to 0.5 Gy protons was characterized at 24 h after exposure by the down-regulation of multiple stress genes (Appendix A). By lowering the proton dose to 0.25 Gy, only several genes were down-regulated (Appendix A), while the other investigated stress genes did not have significant expression changes, indicating a dose-dependent stress response.

Exposure of monocytes to UVB alone and their co-exposure to protons and UVB induced the over-expression of the inflammatory genes *IL1* and *IL1B*, while protons alone induced the down-regulation of these genes in CRL 9855 monocytes (Figure 7c). Moreover, it was found that UVB and the co-exposure of monocytes to UVB and protons did not alter the expression of the pro-inflammatory genes *IL6* and *CXCL8*, while protons strongly down-regulated these genes at 24 h post-exposure (Figure 7c). Altogether, results suggest that cellular stress inflicted by the co-exposure of CRL 9855 monocytes to 0.5 Gy protons and 50 J/m^2^ UVB was largely governed by the UVB exposure, as seen in the viability and stress gene expression changes. Fluence increase above 50 J/m^2^ was shown to elicit a strong and broad stress response. Of note is that the stress response of monocytes was enhanced following fluence-effect curves when cells were exposed to UVB fluences higher than 50 J/m^2^ (Appendix A). Even if some genes were not over-expressed at 50 J/m^2^, they became significantly over-expressed at UVB fluences of 100 and 200 J/m^2^. A tremendous increase in the FC value was registered in the case of the pro-inflammatory *IL6* gene (50 J/m^2^: FC = 0.75 ± 0.13; 100 J/m^2^: FC = 181 ± 5; and 200 J/m^2^: FC = 320 ± 292) (for additional information see also Appendix A).

A briefing of the main results is presented in Table 1.

## 4. Discussion

While the impact of UV radiation, either UVA or UVB, on skin cells has been extensively investigated [49], the potential interference of SCR and solar emissions with the biologic effects of UVB is not known. In our in vitro study, a systems medicine approach [50] was applied by investigating in parallel various types of skin and blood cells, and various components of their response to aggression. The selected doses of protons or gamma rays, and of UVB radiation used in this study did not have important effects on the viability of fibroblasts and keratinocytes cell lines used in this study. While proton or gamma irradiation had only minor effects, UVB decreased more, but not dramatically, the number of metabolically active cells in the investigated fluence domain. From the examined cell lines, the most sensitive cells to UVB were monocytes, followed by keratinocytes, while the less sensitive cells being fibroblasts. The co-exposure of normal cells to protons and UVB exerted a synergistic effect on fibroblasts in terms of viability and DNA damage, with protons intensifying the biological effect of UVB. Meanwhile, the cytotoxic effects of UVB were dominant in keratinocytes and monocytes. While the results obtained from individual cell lines provide valuable insights into cellular responses, it is important to acknowledge that some variability may exist among different cells of the same type due to genetic and metabolic differences. Additionally, we do not rule out the differences in the responses of primary normal human cell lines and primary skin cells. The study by D’Errico et al. investigated the differential sensitivity of human skin cells to UVB radiation, focusing on primary keratinocytes and fibroblasts. The findings reveal that keratinocytes exhibited higher resistance to UVB-induced lethality compared to fibroblasts, as evidenced by higher colony-forming ability post-exposure. However, keratinocytes were more prone to undergo apoptosis following UVB exposure, with a significant increase in apoptotic cells observed at 24 and 72 h post-irradiation. In contrast, fibroblasts showed minimal apoptotic response under similar conditions. These results suggest that keratinocytes employ apoptosis as a protective mechanism to maintain genomic integrity upon UVB-induced DNA damage, whereas fibroblasts exhibit higher sensitivity to UVB-induced cell death, likely due to less efficient activation of apoptotic pathways [51]. A pronounced viability decrease was induced at 48 h after co-exposure of Hs27 cells to gamma rays and UVB as compared to gamma rays alone, indicating a synergistic and persistent impact of gamma rays and UVB in the co-exposure scenario. On the other hand, co-exposure of HaCaT cells to gamma rays and UVB did not induce viability changes at 24 h and 48 h compared to gamma rays or UVB alone. It is worth mentioning that the assay used for assessing cellular viability was the metabolic reduction of tetrazolium salts (MTT/MTS) to formazan (MTT/MTS assay), which is not able to quantify early apoptosis, and thus an overestimation of viability might have occurred [52]. The survival of VH10 fibroblasts was assessed by manually counting with the counting chamber. UVB irradiation at a dose of 100 J/m^2^ significantly reduced the survival rate of VH10 cells, while gamma radiation at 0.5 Gy resulted in a higher survival rate. Notably, the combination treatment of UVB (100 J/m^2^) and gamma radiation (0.5 Gy) further reduced cell survival, indicating a synergistic effect of the mixed exposures (additional information can be found in the Appendix A). These findings suggest a synergistic and persistent impact of combined gamma and UVB radiation on cell viability, particularly in Hs27 and VH10 fibroblasts, while HaCaT cells appear more resistant.

The synergistic action of protons and UVB was best evidenced by the significantly enhanced DNA damages inflicted to normal skin and blood cells lines that decreased over 24 h post-irradiation due to DNA repair mechanisms. Over-expression of some stress genes related to DNA damage further confirmed that DNA damage and repair was still active at 48 h in fibroblasts and keratinocytes. Meanwhile, down-regulation of other stress genes indicated that some repair mechanisms were on their way to resolution. DNA damage was persistent, indicating that cells are affected in the long run, even if they are exposed to relatively low radiation doses of protons and UVB. These results are also aligned with the results from the co-exposure of Hs27 and VH10 fibroblasts to gamma rays and UVB. It is worth mentioning that although comparisons between different cell lines with the same tissue origin provide useful insights, it is important to consider that they originate from different donors, and inter-individual variability in metabolic pathways may influence their responses to radiation exposure. HaCaT keratinocytes have proven to have better repair capacity against UVB, especially when UVB was combined with gamma rays. Keratinocytes, the primary cells of the epidermis, are more susceptible to apoptosis following UVB exposure, primarily due to their role in protecting deeper layers of the skin. This apoptotic response is mediated by the activation of p53, a key regulator of the DDR which drives cells with severe damage towards programmed cell death. In contrast, dermal fibroblasts exhibit a more robust DNA repair capacity and are less prone to apoptosis under similar conditions. This distinction underscores the specialized roles of these cells: keratinocytes prioritize removing damaged cells to prevent mutagenesis, while fibroblasts emphasize survival and repair to maintain dermal integrity [52]. Moreover, results show that cells exposed first to UVB, and 20 min thereafter, to proton beams, appeared to be affected more than cells exposed first to protons and then to UVB in terms of DNA damage. While proton-first exposure followed by UVB resulted in more immediate and sustained damage, UVB-first exposure followed by protons was leading to an amplified damage response.

Significant differences were observed in the percentages of colocalization of γH2AX and 53BP1 under varying exposure conditions, providing insights into the cellular response to DNA damage. High percentages of colocalization were observed in fibroblasts exposed to 0.5 Gy gamma radiation. These findings align with the well-established role of γH2AX and 53BP1 in double-strand break repair following ionizing radiation, as highlighted by Chaurasia et al. [53], where discrete γH2AX foci colocalize with 53BP1 as part of the DNA repair machinery. In contrast, exposure to 100 J/m^2^ UVB resulted in significantly lower colocalization levels, consistent with the study by Marti et al. [17] demonstrating that that γH2AX phosphorylation induced by UVB is primarily pan-nuclear and is associated with NER, rather than being localized to discrete sites of double-strand breaks where 53BP1 typically accumulates. Therefore, the reduced colocalization of γH2AX with 53BP1, a marker for DSBs, aligns with the study’s observation that γH2AX in S-phase cells post-UV exposure does not correspond to the extent of DNA damage in the form of DSBs. Accordingly, the γH2AX signal in the 100 J/m^2^ UVB exposure may be indicative of replication stress rather than direct DNA damage. This is supported by the conclusion of Dhuppar et al. that γH2AX accumulation in S-phase cells after UV irradiation corresponds to DNA replication activities and not directly to DNA damage levels [54]. Interestingly, the co-exposure results show that colocalization of γH2AX and 53BP1 foci is significantly influenced by the type and combination of exposures, reflecting the complexity of DNA damage. After 0.5 Gy exposure, high colocalization at 1 h indicates predominantly simple DSBs that are typical of low-LET radiation and are efficiently recognized and repaired over time. In contrast, co-exposure to 0.5 Gy protons and 100 J/m^2^ UVB radiation resulted in statistically significant lower colocalization at all time points, highlighting the generation of more complex lesions. This aligns with findings from previous studies [55], which demonstrate that clustered DNA damage or overlapping lesion types reduce colocalization due to impaired recruitment of repair protein. The persistence of lower colocalization in the co-exposure condition suggests the involvement of challenging-to-repair lesions, such as UVB-induced photoproducts combined with radiation-induced DSBs, requiring overlapping repair pathways.

To corroborate the observed higher levels of γH2AX in fibroblasts co-exposed to 0.5 Gy gamma rays and 100 J/m^2^ UVB, TEM analysis provided ultrastructural evidence supporting the enhanced DNA damage response in co-exposed samples. Specifically, TEM revealed a markedly higher number of γH2AX particles in the nuclei of co-exposed fibroblasts. These particles also seemed prominently localized to areas of chromatin condensation, consistent with sites of DNA damage and repair activity. Furthermore, in co-exposed samples, a significantly higher number of γH2AX particle clusters was observed, indicating the aggregation of multiple DNA damage events within close regions of the nucleus. The increased density and clustering of γH2AX particles in co-exposed samples align with findings obtained throughout the IF assay and reinforce the hypothesis of a synergistic interaction between gamma rays and UVB radiation. The combined exposure likely amplifies the recruitment and retention of γH2AX at sites of complex DNA damage, where DSBs and UVB-induced lesions coexist. This synergistic effect may reflect the simultaneous activation of distinct repair pathways, such as homologous recombination and NER, within the same cellular environment.

UVB irradiation and gamma radiation, both individually and in combination, significantly induce cellular senescence in VH10 fibroblast cells. Control cells displayed a low baseline level of senescence, which increased markedly following UVB irradiation and gamma irradiation. The combined treatment of UVB and gamma radiation caused an even higher increase in senescence, and a highly significant difference compared to both the control and single exposures, highlighting the synergistic effect of UVB and gamma radiation. These results align with the observations of increased DNA damage and sustained pan-nuclear γH2AX staining under combined exposure conditions in Hs27 fibroblasts. The substantial elevation in senescence levels following mixed radiation exposure of VH10 cells, presented in Appendix A, underscores the heightened impact of these combined treatments on cellular aging and stress responses. The overexpression of the *CDKN1a* gene in co-exposed Hs27 fibroblasts keratinocytes indicated that these cells might be committed to senescence [56] and cell cycle arrest [57] required for efficient DNA damage repair in aggressed cells. The senescent phenotype is sustained also the by the over-expression of *IL1* genes in co-exposed cells, it being known that *IL-1* controls the late arm of the senescence secretome induced by *NF-κB* and has a pro-tumorigenic action [58].

Chromosomal aberration assay revealed distinct responses in lymphocytes and fibroblasts under varying radiation exposure conditions. Dicentric yields in UVB-exposed lymphocytes were not significantly different from the background levels in unexposed cells, but a significant induction of dicentrics was observed in cells exposed to 0.5 Gy protons and in all combined exposures to protons and UVB. Despite the use of only two donors due to organizational constraints, the dicentric assay remains a reliable biomarker for radiation exposure, with our results supported by in vitro calibration data, replicate cultures, and comparable dicentric induction between donors (*p* < 0.005). However, future studies should include more donors to account for inter-individual variability. Similarly, gamma irradiation demonstrated a synergistic enhancement in chromosomal aberration yields when combined with UVB exposure in Hs27 fibroblasts. These findings align with the observed colocalization patterns of γH2AX and 53BP1 in the same exposure conditions and together, highlighting the complex interplay between different radiation types. Of note is that gamma rays and UVB synergistically amplify both chromosomal aberrations and the colocalization of key DNA damage markers, reflecting a heightened and coordinated DNA damage response. Lambert et al. found a two-fold increase in the frequency of dicentric chromosomes in G0 lymphocytes after combined UVC (19.6 J/m^2^) and X-ray (1.5 Gy) irradiation. They hypothesized that this synergistic effect of UV and X-ray irradiation on the yield of dicentric chromosomes might be due partly to the utilization of the same repair enzymes during healing of X-ray and UV-induced DNA lesions. In contrast, Holmberg did not detect an increase in the dicentrics yield in G1-lymphocytes after combined irradiation with UV and X-rays. They observed the induction of a low, but not significant, dicentric frequency (10 dicentrics in 1000 metaphases) in samples of two different donors out of three donors analyzed after irradiation of lymphocytes in the G1 stage with 7.5 J/m^2^ UV radiation. However, UV exposure of unstimulated G0 lymphocytes induced a low frequency (2–3%) of dicentric chromosomes in addition to chromatid-type aberrations of the order of 5–10% [59,60]. This synergistic effect of mixed irradiation with regard to the induction of various chromosomal aberration types and consequences, respectively, e.g., micronuclei and DNA repair foci, has been described in the literature several times [61,62,63,64,65,66]. Just recently, Lopéz Riego et al. found a higher chromosomal aberration frequency after mixed exposures than expected in both donors after irradiation of peripheral blood lymphocytes from two donors with X-rays (0 to 2 Gy), alpha particles (0 to 2 Gy), or both radiation qualities (1:1 mixture, each half the dose). However, structural aberrations such as dicentric chromosomes could not be analyzed due to poor quality of chromosome spreads. Therefore, the total number of chromosomes and fragments per metaphase served as the basis for the assessed aberration frequency. The use of sister chromatid exchange (SCE) as a biomarker for assessing human exposure to environmental mutagens is extensively studied by Revel [67]. The author emphasizes its potential in identifying populations at risk due to environmental mutagen exposure as SCEs involve the reciprocal exchange of DNA between sister chromatids during cell division and can be induced by various chemical and physical agents. The absence of chromatid exchanges in the gamma-ray-only samples could imply efficient repair of DSBs without significant misrepair or interchromosomal recombination under these experimental conditions. UVB radiation, which is non-ionizing, primarily induces DNA damage in the form of CPDs and 6-4 photoproducts, leading to SSBs and localized lesions. The low frequency of chromatid exchanges in the UVB-exposed samples could reflect the localized and less severe nature of UVB-induced damage compared to the widespread impact of gamma rays. The combination of gamma rays and UVB likely induced a complex spectrum of DNA damage, including DSBs, SSBs, and bulky DNA adducts. The interaction between these damage types may overwhelm the cellular repair machinery, leading to misrepair or incomplete repair processes. This could explain the increased frequency of chromatid exchanges in co-exposed cells, a hallmark of genomic instability and recombinational events. Co-exposure may amplify oxidative stress, disrupt cell-cycle checkpoints, and impair DNA repair pathways. For instance, UVB-induced lesions may interfere with the processing or repair of gamma-ray-induced DSBs, increasing the likelihood of chromosomal aberrations. Chromatid exchanges are indicative of severe genomic instability. Kuznetsov et al. in a study that focuses on key cellular components, such as chromatin structure, membrane permeability, mitochondrial activity, and calcium ion distribution, found that both ionizing (X-rays and gamma rays) and non-ionizing radiations (high-frequency microwaves), individual and in combination, can alter chromatin condensation, increase membrane permeability, inhibit mitochondrial function, and disrupt calcium ion balance within cells. These results highlight the complex nature of cellular responses to combined physical stressors and underscore the need for further research to understand the potential health implications of such combined exposures [68].

The enhanced harmful effects of the co-exposure of cells to protons and UVB were further demonstrated by a transcriptomic study on 84 stress genes assessed relatively late after exposure.

The observed over-expression of death genes indicated that adhered fibroblasts and kerationocytes, presumably living cells, were committed to increased cell death, indicating that the stress response was not able to repair the damages inflicted by combined proton and UVB radiations. A more complex stress response was developed by co-exposed fibroblasts and keratinocytes, as compared to cells exposed to single radiations, combining common and distinctive over-expressed genes involved in the DNA damage response, cell death, and oxidative and hypoxic stress. As such, co-exposed cells appear to be more stressed than those exposed either to protons or to UVB. The magnitude of the expression changes of some stress genes in co-exposed cells was the same as in cells exposed to single radiations, suggesting that the intensity of the stress response could not increase over a certain threshold, hence limiting the repair ability of co-exposed cells facing multiple cues. This behavior may account for the increased DNA lesions registered in co-exposed cells, translated in increased cell death and genomic instability. Additionally, several inflammation genes were found up-regulated in co-exposed immune and non-immune cells, such as the *IL1* genes whose products can act in an autocrine and paracrine manner [69]. Inflammation is a powerful repair mechanism in the skin, but it may lead to disease if it becomes chronic [70,71].

The gene expression pattern and the viability changes in co-exposed monocytes highlighted that their behavior was ruled by the exposure to 50 J/m^2^ UVB. The expression of several stress genes followed a dose–effect relationship in monocytes, indicating that their response to DNA damage, proteotoxicity, and oxidative and hypoxic stress could be even stronger at higher UVB fluences of 100–200 J/m^2^.

As certain stress genes were found down-regulated, especially in monocytes, it appears that the stress response was partly on its way to extinguish. Nevertheless, cells still remained under stress late after exposure, although the investigated proton doses and UVB fluences were relatively low (0.5 Gy, and 50 J/m^2^, respectively). In the real world, if cells are co-exposed chronically to SCR and UVB, more damaging health effects are expected in the long run than those inflicted by solitary radiation hits [72,73].

Our research gives important information regarding the complicated interaction between ionizing and UVB radiations in human normal cells, presenting possible hazards when combined exposures occur. The synergistic action is manifested by the persistent DNA damage, changed expression of stress genes, and elevated chromosomal instability exceeding in most cases the pure additive action of radiations. These findings emphasize the importance of determining damage thresholds at which cellular repair mechanisms are overwhelmed, leading to genomic instability over long periods. By simulating conditions relevant to ozone depletion and increased secondary cosmic radiation exposure, our research presents a model for anticipating potential health risks in extreme environmental situations. It is necessary to understand these mechanisms in order to create protective measures and improve risk assessment models for people exposed to such conditions. Elaborating on these observations, examining inter-individual variability, and assessing the long-term biological impact of combined radiation exposure should be the aim of future research.

## 5. Conclusions

A prolonged stress was evidenced in human non-malignant keratinocytes, fibroblasts, and monocytes exposed in vitro to combined protons or gamma rays and UVB radiation being a mimic of an increased exposure of humans to combined radiation fields reaching the Earth due to a potential progressive depletion of the stratospheric ozone layer. Co-exposure of cells inflicted more profound and persistent DNA damages than either radiation alone, and various repair mechanisms were identified related to DDR, inflammation, antioxidant, and hypoxia responses.

## Figures and Tables

**Figure 1 biomolecules-15-00536-f001:**
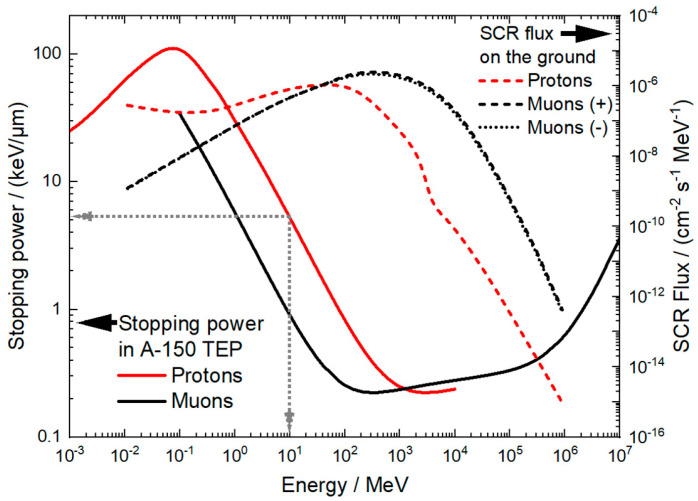
Stopping power of protons and muons in A-150 tissue-equivalent plastic (data from NIST pstar: https://physics.nist.gov/cgi-bin/Star/ap_table.pl, accessed on 1 January 2025, and Groom, D.E., Mokhov, N.V., Striganov, S.I., 2001. Muon stopping power and range tables 10 MeV–100 TeV. At. Data Nucl. Data Tables 78, 183–356 [23]), and typical secondary cosmic radiation (SCR) fluxes in the environment [22].

**Figure 2 biomolecules-15-00536-f002:**
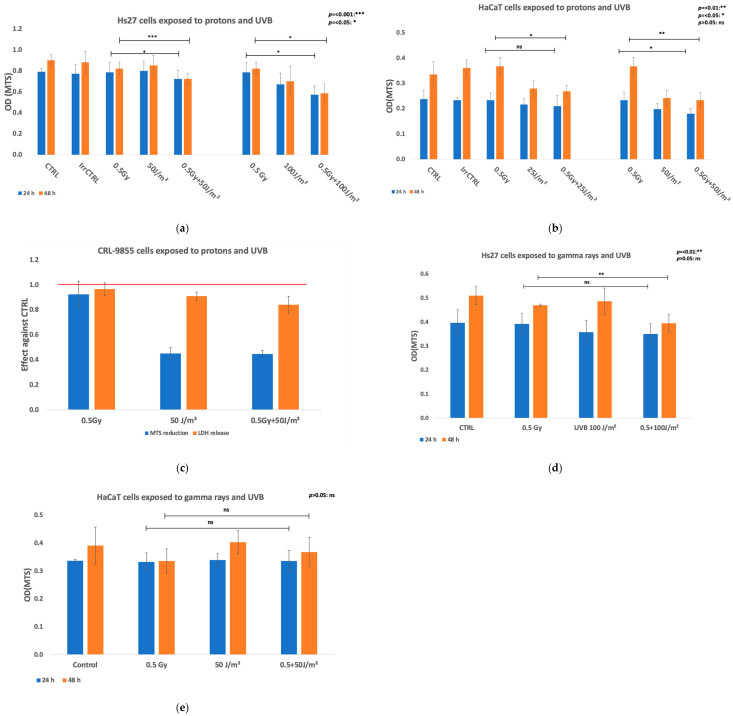
Cellular viability measurements in human non-malignant cells. (**a**) MTS reduction by Hs27 fibroblasts at 24 h and 48 h after exposure to 0.5 Gy protons, and thereafter, within approx. 20 min, to UVB (50 J/m^2^ or 100 J/m^2^) (mean effect ± SD for 3 independent experiments); (**b**) MTS reduction by human normal HaCaT keratinocytes after exposure to 0.5 Gy protons, and thereafter, within approx. 20 min, to UVB (25 J/m^2^ or 50 J/m^2^) 24 h and 48 h post-exposure (mean OD ± SD for 3 independent experiments); and (**c**) MTS reduction and LDH release by human normal CRL 9855 monocytes at 24 h after exposure to 0.5 Gy protons, and, within approx. 20 min thereafter, to 50 J/m^2^ UVB. The effect of co-exposure was calculated as OD of co-exposed samples divided by the mean OD value of unexposed control cells (CTRL). The red line designates the unit effect (no effect); (**d**) MTS reduction by Hs27 fibroblasts at 24 h and 48 h after exposure to 0.5 Gy gamma rays, and thereafter, within approx. 20 min, to UVB (100 J/m^2^) (mean effect ± SD for 2 independent experiments); and (**e**) MTS reduction by human normal HaCaT keratinocytes after exposure to 0.5 Gy gamma rays, and thereafter, within approx. 20 min, to UVB (50 J/m^2^), analyzed at 24 h and 48 h post-exposure (mean OD ± SD for 2 independent experiments).

**Figure 3 biomolecules-15-00536-f003:**
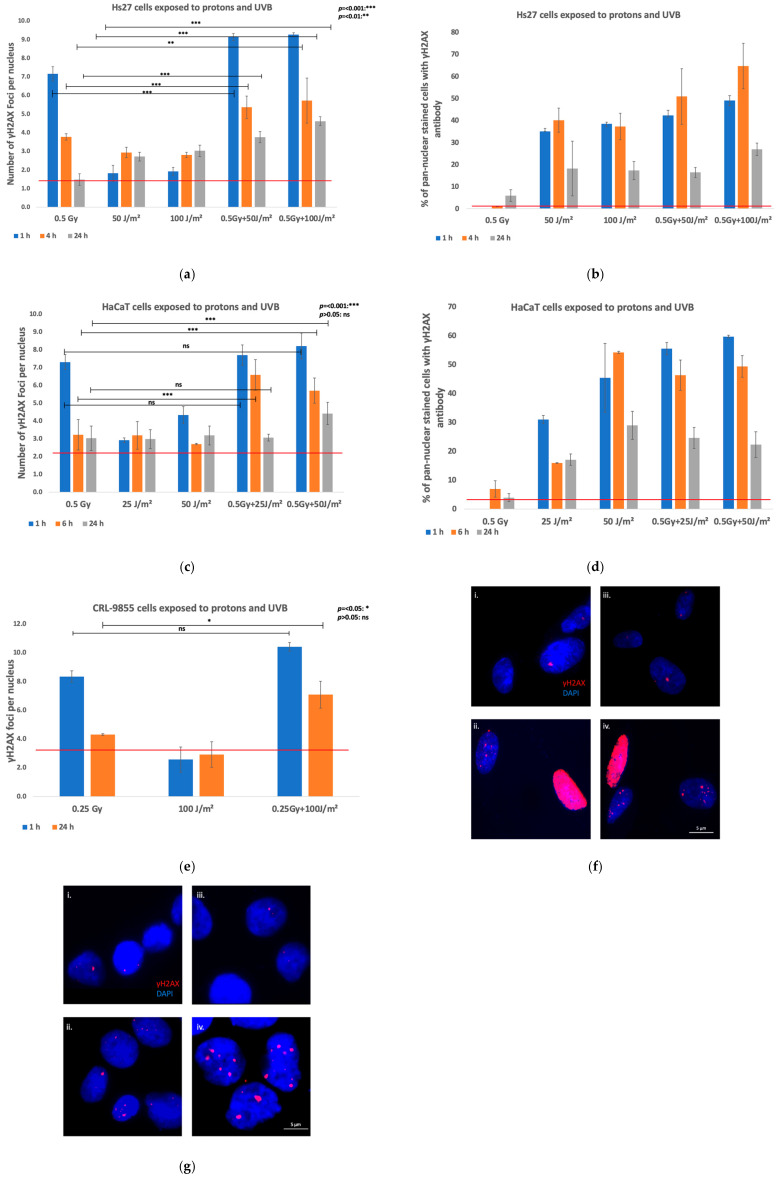
γH2AΧ staining in human non-malignant cells following co-exposure to protons and UVB. Results are presented as mean effect (value) ± SD for two independent experiments. The red line designates the mean value in control samples. (**a**) γH2AΧ foci per nucleus in Hs27 cels at 1 h, 4 h, and 24 h after exposure to 0.5 Gy protons or UVB (50 J/m^2^ or 100 J/m^2^) as single stressors, or to combined challenges (protons first, and UVB 20 min thereafter) (stdev_control_ = 0.17 foci/nucleus); (**b**). Pan-nuclear γH2AΧ staining in Hs27 cells at 1 h, 4 h, and 24 h after exposure to 0.5 Gy protons or UVB (50 J/m^2^ or 100 J/m^2^) as single stressors, or to combined challenges (protons first, and UVB 20 min thereafter) (stdev_control_ = 0.99%); (**c**) γH2AΧ foci per nucleus in HaCaT cells at 1 h, 6 h, and 24 h after exposure to 0.5 Gy protons or UVB (25 J/m^2^ or 50 J/m^2^) as single stressors, or to combined challenges (protons first, and UVB 20 min thereafter) (stdev_control_ = 0.37 foci/nucleus); (**d**) pan-nuclear γH2AΧ staining in HaCaT cells at 1 h, 6 h, and 24 h after exposure to 0.5 Gy protons or UVB (25 J/m^2^ or 50 J/m^2^) as single stressors, or to combined challenges (protons first, and UVB 20 min thereafter) (stdev_control_ = 3.35%); (**e**) γH2AΧ foci per nucleus in CRL 9855 at 1 h and 24 h after exposure to 0.25 Gy protons or UVB (100 J/m^2^) as single stressors, or to combined challenges (protons first, and UVB 20 min thereafter) (stdev_control_ = 0.26 foci/nucleus); (**f**) fluorescence images of Hs27 cells following co-exposure to protons and UVB: (i) unirradiated, (ii) exposed to 50 J/m^2^ UVB, (iii) exposed to 0.5 Gy protons, and (iv) co-exposed to 0.5 Gy protons and 100 J/m^2^ UVB all 24 h post-exposure (blue signal: cell nucleus, red signal: γH2AΧ foci); and (**g**) fluorescence images of HaCaT cells following co-exposure to protons and UVB: (i) unirradiated, (ii) exposed to 50 J/m^2^ UVB, (iii) exposed to 0.5 Gy protons, and (iv) co-exposed to 0.5 Gy protons and 50 J/m^2^ UVB all 24 h post-exposure (blue signal: cell nucleus, red signal: γH2AΧ foci). Scale bar is indicative of the nucleus size.

**Figure 4 biomolecules-15-00536-f004:**
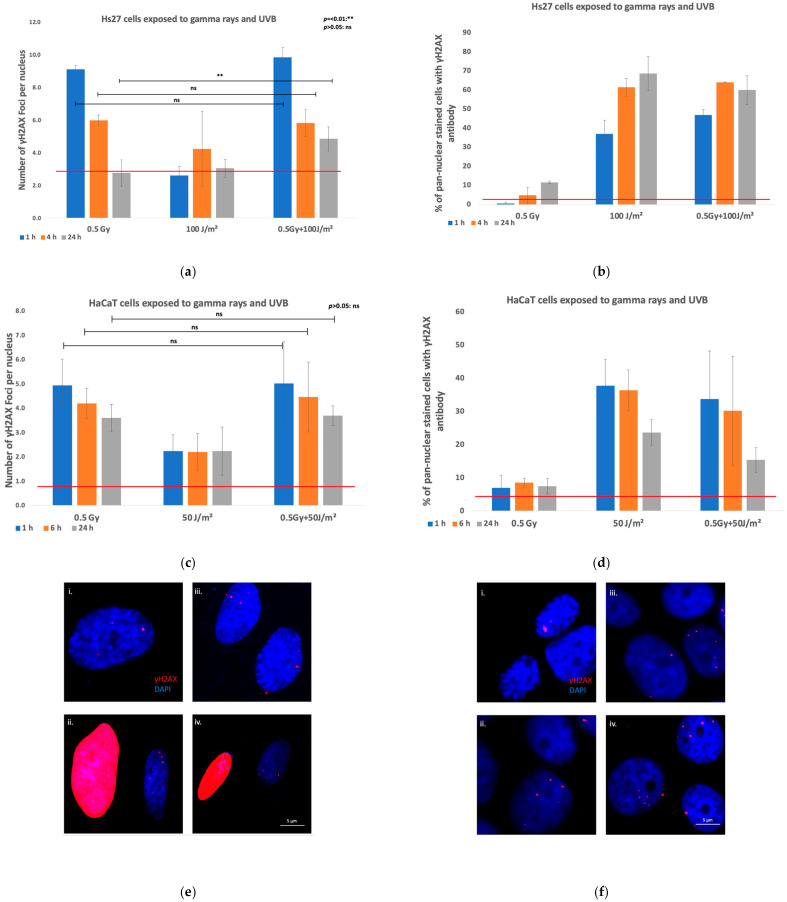
γH2AΧ staining in human non-malignant cells following co-exposure to gamma rays and UVB. Results are presented as mean effect (value) ± SD for two independent experiments. The red line designates the mean value in control samples. Scale bar is indicative of the nucleus size. (**a**) γH2AΧ foci per nucleus in Hs27 cells at 1 h, 4 h, and 24 h after exposure to 0.5 Gy gamma rays or 100 J/m^2^ UVB as single stressors, or to combined challenges (protons first, and UVB 20 min thereafter) (stdev_control_ = 0.45 foci/nucleus); (**b**) pan-nuclear γH2AΧ staining in Hs27 cells at 1 h, 4 h and 24 h after exposure to 0.5 Gy gamma rays or UVB 100 J/m^2^ as single stressors, or to combined challenges (gamma rays first, and UVB 20 min thereafter) (stdev_control_ = 3.47%); (**c**) γH2AΧ foci per nucleus in HaCaT cells at 1 h, 6 h, and 24 h after exposure to 0.5 Gy gamma rays or 50 J/m^2^ UVB as single stressors, or to combined challenges (gamma rays first, and UVB 15 min thereafter) (stdev_control_ = 0.09 foci/nucleus); (**d**) pan-nuclear γH2AΧ staining in HaCaT cells at 1 h, 6 h, and 24 h after exposure to 0.5 Gy protons or 50 J/m^2^ UVB as single stressors, or to combined challenges (gamma rays first, and UVB 20 min thereafter) (stdev_control_ = 1.14%); (**e**) fluorescence images of Hs27 cells following co-exposure to gamma rays and UVB: (i) unirradiated, (ii) exposed to 100 J/m^2^ UVB, (iii) exposed to 0.5 Gy gamma rays, and (iv) co-exposed to 0.5 Gy gamma rays and 100 J/m^2^ UVB all 24 h post-exposure (blue signal: cell nucleus, red signal: γH2AΧ foci); and (**f**) fluorescence images of HaCaT cells following co-exposure to gamma rays and UVB: (i) unirradiated, (ii) exposed to 50 J/m^2^ UVB, (iii) exposed to 0.5 Gy gamma rays, and (iv) co-exposed to 0.5 Gy protons and 50 J/m^2^ UVB all 24 h post-exposure (blue signal: cell nucleus, red signal: γH2AΧ foci).

**Figure 5 biomolecules-15-00536-f005:**
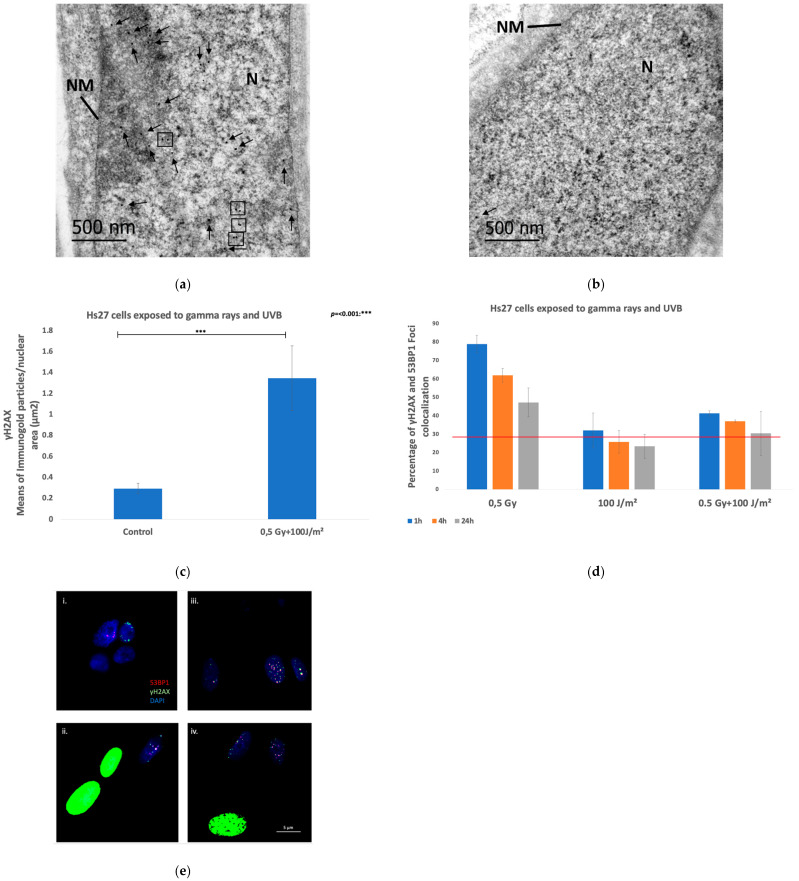
Complex DNA damage detection. The red line designates the mean value in control samples. (**a**) Representative electron micrographs of γH2AX immunogold localization in the nuclei of Hs27 fibroblasts co-exposed to 100 J/m^2^ and 0.5 Gy of gamma rays processed 24 h post-irradiation. Cell arrows indicate single immunogold particles (15 nm gold) and boxes indicate immunogold particles in very close proximity (clusters); (**b**) representative electron micrographs of γH2AX immunogold localization in the nuclei of non-irradiated Hs27 fibroblasts processed 24 h post-irradiation. Cell arrows indicate single immunogold particles (15nm gold) and boxes indicate immunogold particles in very close proximity (clusters); (**c**) number of DSBs induced by co-exposure to 100 J/m^2^ and 0.5 Gy of gamma rays 24 h post-irradiation. γH2AX marker was used for the detection of DSBs and analysis was performed by TEM analysis and Image J. Results indicate mean number of γH2AX particles per nuclear area μm^2^. N: nucleus, NM: nuclear membrane. Scale bar: 500 nm; original magnification: 20,000×; (**d**) colocalization percentages γH2AΧ and 53BP1 staining in normal Hs27 fibroblasts at 1 h, 4 h, and 24 h after exposure to 0.5 Gy gamma rays or 100 J/m^2^ UVB as single stressors, or to combined challenges. Results are presented as mean effect (value) ± SD for two independent experiments. (stdev_control_ = 2.01%); (**e**) fluorescence images of Hs27 cells (i) unirradiated, (ii) exposed to 100 J/m^2^ UVB, (iii) exposed to 0.5 Gy gamma rays, and (iv) co-exposed to 0.5 Gy gamma rays and 100 J/m^2^ UVB all 24 h post-exposure (blue signal: cell nucleus, green signal: γH2AΧ foci, and red signal: 53BP1 foci). Scale bar is indicative of the nucleus size.

**Figure 6 biomolecules-15-00536-f006:**
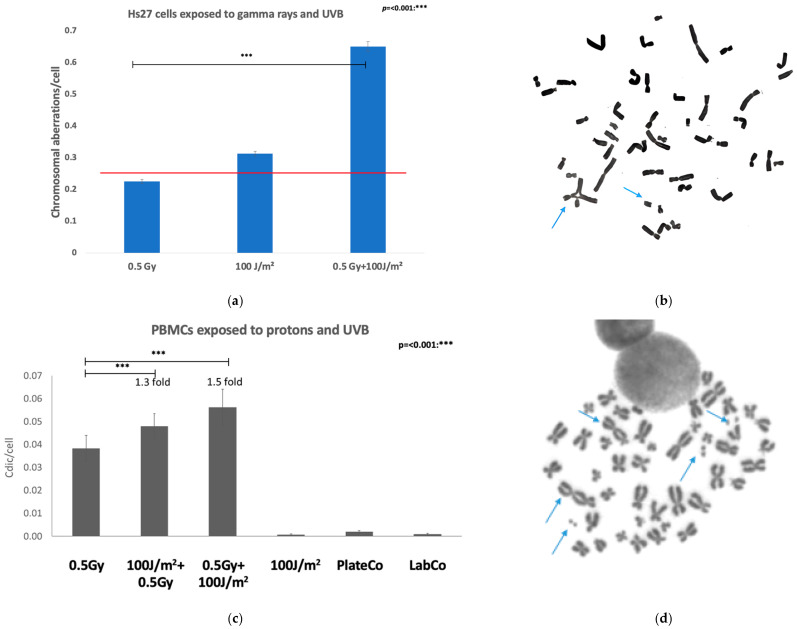
Chromosomal aberration in non-malignant cells. (**a**) Chromosomal aberration frequencies in HS27 cells induced by 0.5 Gy gamma rays or 100 J/m^2^ UVB as single stressors, or to combined challenges. Results are presented as mean effect (value) ± SD for two independent experiments. The red line designates the mean value in control samples (stdev_control_ = 0.006 CA/cell); (**b**) metaphase spread of Hs27 cells exposed to 0.5 Gy gamma rays and 100 J/m^2^ UVB with two chromosomal aberrations (blue arrows: chromosomal exchange and acentric fragment); (**c**) dicentric frequencies in PBMCs induced by 0.5 Gy protons, 100 J/m^2^ UVB or 400 J/m^2^ UVB, combined exposures (protons + UVB and vice versa), neutron–gamma background (plate control) as well as observed in the PTB laboratory control; and (**d**) PBMC metaphase spread exposed to 0.5 Gy gamma rays and 100 J/m^2^ UVB with two dicentric chromosomes and three acentric fragments.

**Figure 7 biomolecules-15-00536-f007:**
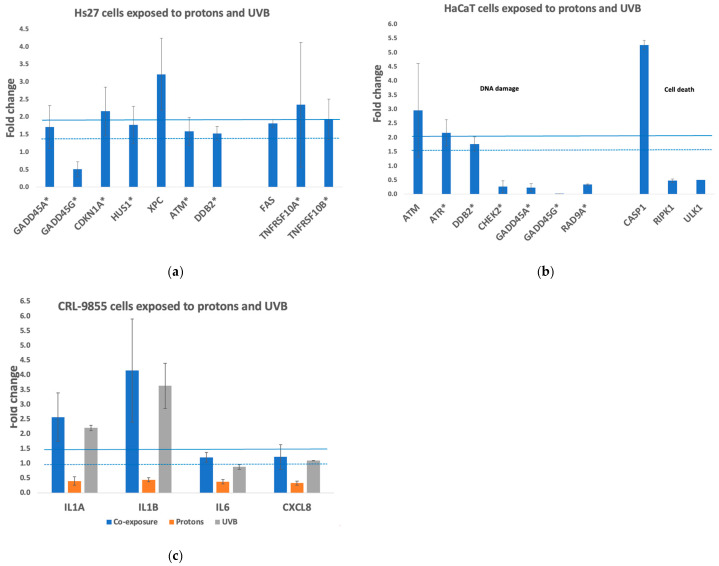
Stress genes with modified expression in non-malignant human cells exposed to protons or UVB applied as single stressors or combined (proton irradiation followed by UVB exposure within approx. 20 min). Genes with mean fold change > 1.5 (gene up-regulation) or <0.7 (gene down-regulation) were represented as mean value ± SD in three independent experiments. Genes with a significant fold change value in two out of three experiments and no significant effect in the third experiment were marked with *. The blue line is the low threshold for significant gene up-regulation, while the dashed blue line corresponds to “the no effect” case for evidencing gene down-regulation. (**a**) DNA damage and repair genes and cell death genes in Hs27 fibroblasts co-exposed to 0.5 Gy protons and 50 J/m^2^ UVB; (**b**) DNA damage and repair genes and cell death genes in HaCaT fibroblasts co-exposed to 0.5 Gy protons and 50 J/m^2^ UVB (mean FC value ± SD in three independent experiments); and (**c**) inflammation genes with modified expression in human normal CRL 9855 monocytes exposed to 0.5 Gy protons or 50 J/m^2^ UVB applied as single stressors or combined (proton irradiation followed by UVB exposure within 20 min). Results are presented as mean value ± SD in two independent experiments.

**Table 1 biomolecules-15-00536-t001:** Briefing of results obtained through different assays.

Assay	Exposure Set Up	Cell Type	Results
Viability (MTS reduction)	Protons + UVB	Keratinocytes (HaCaT)	Significantly lower in co-exposed cells at 24 h but at 48 h was mainly due to UVB exposure.
		Fibroblasts (Hs27)	Marked decrease in MTS reduction at 24 h and 48 h after irradiation in co-exposed cells both against protons and UVB alone.
		Monocytes (CRL-9855)	Marked decrease in MTS reduction at 24 h post-exposure was due to UVB exposure.
	Gamma rays + UVB	Keratinocytes (HaCaT)	MTS reduction did not alter at 24 h or 48 h post-co-exposure.
		Fibroblasts (Hs27)	Pronounced decrease at 48 h after co-exposure.
Viability (manual counting)	Gamma rays + UVB	Fibroblasts (VH10)	Co-exposure reduced cell survival.
DNA Damage IF γH2AΧ	Protons + UVB	Keratinocytes (HaCaT)	In co-exposed cells, the γH2AX foci number increased 6 h post-exposure (particularly in the co-exposure with the higher UVB dose). This synergistic effect was still evident at 24 h.
		Fibroblasts (Hs27)	Co-exposed with the higher UVB fluence displayed markedly higher levels of γH2AX foci per nucleus at all time points.
		Monocytes (CRL-9855)	In co-exposed, the number of γH2AX foci was increased, the synergistic effect persisting over 24 h.
	Gamma rays + UVB	Keratinocytes (HaCaT)	The co-exposure to did not result in statistically higher number of γH2AX foci.
		Fibroblasts (Hs27)	The co-exposure resulted in more severe damage, with a statistically significant difference at the 24 h time point.
		Fibroblasts (VH10)	The co-exposure to resulted in significantly greater damage across all time points.
DNA Damage IF γH2AΧ-53PB1	Gamma rays + UVB	Fibroblasts (Hs27)	Co-exposure resulted in high percentages of colocalization compared to UVB-exposed samples, but significant lower at all time points compared to samples exposed to ionizing radiation.
Transmission electron microscopy—γH2AX immunogold labeling	Gamma rays + UVB	Fibroblasts (Hs27)	In co-exposed cells, the number of γH2AX particles per μm^2^ of nuclear surface area was significantly elevated compared to the control sample while 48.75% of the analyzed nuclear areas exhibited γH2AX gold particle clusters.
Genomic instability-Chromosomal aberrations	Protons + UVB	PBMCs	Dicentric yield increased about 1.5-fold in co-exposed cells compared to proton-irradiated samples.
	Gamma rays + UVB	Fibroblasts (Hs27)	The chromosomal aberration yields in co-exposed cells was increased about 1.5-fold compared to gamma irradiated samples.
Gene expression	Protons + UVB	Keratinocytes (HaCaT)	*ATM*, *ATR*, *DDB2*, and *CASP1* were up-regulated at 48 h post co-exposure.
		Fibroblasts (Hs27)	*GADD45A*, *CDKN1A*, *HUS 1*, *XPC*, *ATM DDB2*, *FAS*, and *TNFRSF10* were up-regulated at 48 h post co-exposure.
		Monocytes (CRL-9855)	The *TNFRSF1* gene involved in necrotic cell death, was down-regulated only in co-exposed cells, while single stressors did not significantly affect their expression. Only the pro-inflammatory stress genes IL1Aand IL1B were up-regulated at 24 h in co-exposed cells or in cells exposed to UVB alone, whilst being down-regulated in cells exposed to protons alone.
Senescence (SA-β-gal)	Gamma rays + UVB	Fibroblasts (VH10)	The mixed radiation treatment increased the percentage of senescent cells.

## Data Availability

All necessary data has been included in this study. Any requests should be addressed to corresponding author A.G.G.

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
