# Peer review of "Synergistic Effects of UVB and Ionizing Radiation on Human Non-Malignant Cells: Implications for Ozone Depletion and Secondary Cosmic Radiation Exposure"

_biomolecules, 2025, doi:10.3390/biom15040536_

Round 1

Reviewer 1 Report

Comments and Suggestions for Authors

Manuscript ID: biomolecules-3499358
Type of manuscript: Article
Title: Biological effects of UVB radiation and protons or gamma rays on human
non-malignant cells

Review:

In this study the authors have investigated the impact of combining UVB radiation and ionizing radiation on different cell types. In some assays monitoring cellular and DNA alterations the combination caused synergistic effects that were larger than either single type of radiation exposure. The manuscript is a peculiar mixture of interesting and detailed experiments addressing multiple variables along with some proofreading issues and potential statistical significance concerns.

Major items:

  1. Lines 570-573, Figure 2c and lines 579-584, Figure 2d and lines 616-618, Figure 2f -

The authors compare treatment columns to each other and say they are significantly different from each other, indicating p < 0.05 or saying that there is a synergistic effect difference in some cases, but the error bars on the two columns are overlapping with each other. By definition, if the standard deviation error bars overlap between two averages then they are not statistically significantly different from each other. In particular, no effect should be called a synergistic increase or decrease if the error bars are overlapping. I believe that the text descriptions of these effects should be changed to reflect a more accurate interpretation.

  1. Figure 2b - These LDH assay columns have extraordinarily large error bars. It is understood that some types of assays by their nature have more variability and higher standard deviations than other assays, but these are larger than any I have ever seen published. At minimum, these error bars mean that no accurate comparisons can be made because statistical significance is impossible to establish. When researchers get results like this it is common to repeat the experiment using a larger number of samples to improve the statistics or to refine/adjust the methodology used in the assays to try to reduce dispersion or to do both of these things. It is likely that the experiment that generated the graph in Figure 2b should be repeated to produce results that can be interpreted.

  1. Many of the graphs have a prominent red horizontal line that represents the level observed in untreated cells. This may be okay, but it is unusual because we don't get to see the error bars on the untreated samples. We would normally ask if the results with the treated samples are higher than those of the untreated samples and if they are outside the standard deviation seen in the untreated samples. This information is not available when a simple horizontal line is used to represent the untreated controls.

  1. Figure 3b - Should 0.5 Gy have a blue column for the 1h timepoint like the others or a space for it if the value was zero?

Figure 2a and other graphs' labels - Should p < 0.5:* be p < 0.05:*?

Minor items:

line 42 - change "especially on" to "especially in"

line 49 - "on earth and at" --> remove the word "and"

lines 58-61 - the first half of the sentence beginning with "Results..." is unclear

line 97 - after "magnitude" add the word "higher"

line 97 - reference number 7 doesn't appear to be the correct reference for this sentence

All figures - the different parts of the figures show lowercase letters in parentheses below them, but it is more normal to use uppercase letters placed on the upper left of each part. The editors can hopefully provide guidance on what is acceptable.

All figures - it is probably unnecessary to define one, two and three asterisks based on p < .001, .01 and .05 on every graph.

line 110 - change "ground" to "ground level"

line 181 - should it be "0.35 x 106 cells"

lines 584 - 591 and 593 - 599 - it appears that multiple sentences were duplicated accidentally so they appear twice in a row

lines 758, 859 and 937 - Figures 3-g and Figures 4-g and Figures 4h all appear to need adjustment.

Figure 5 legend - the text shows "*: no metaphase cells obtained; symbol: no dicentrics observed" but it's not clear what these symbols are referring to.

Comments on the Quality of English Language

There are a few difficult sentences but it is mostly okay.

Author Response

Response to comments by Reviewer 1:

We would like to thank the reviewer for their valuable feedback and constructive comments on our manuscript. Their insights have greatly helped improve the quality of our work.

Regarding the 1st Major correction “Lines 570-573, Figure 2c and lines 579-584, Figure 2d and lines 616-618, Figure 2f -The authors compare treatment columns to each other and say they are significantly different from each other, indicating p < 0.05 or saying that there is a synergistic effect difference in some cases, but the error bars on the two columns are overlapping with each other. By definition, if the standard deviation error bars overlap between two averages then they are not statistically significantly different from each other. In particular, no effect should be called a synergistic increase or decrease if the error bars are overlapping. I believe that the text descriptions of these effects should be changed to reflect a more accurate interpretation” we provide the following response:

Response: Data were analyzed using Student’s t-Test: Paired Two Sample for Means for comparison. Differences were highlighted clearly between the response of co-exposed HaCaT keratinocytes versus cells exposed solely to protons but not to UVB alone (Figure 2cà Figure 2b)). We corrected the text accordingly: “When the UVB fluence of 50 J/m² was used in the co-exposure setting with 0.5 Gy protons (Figure 2càFigure 2b), MTS reduction by HaCaT keratinocytes was significantly lower in co-exposed samples compared to protons alone both at 24 h (p<0.05) and 48 h (p<0.01), but was comparable to the data registered in samples exposed to UVB alone (Figure 2c), indicating that the effect was mainly due to the UVB exposure. Regarding Figure 2fàFigure 2d which presents the MTS results for Hs27 cell line statistical significant difference was found using Student’s t-Test with p<0.01. Attached we provide the excel sheet: MTS assay Gamma rays+UVB Hs-27.

Indeed, cellular systems at 48 h post-exposure were unstable regarding LDH release in independent experiments both in the case of Hs27 fibroblasts (Figure 2b) and HaCaT keratinocytes (Figure 2d), limiting the significance of results. We reworded accordingly the results presentation as follows: “LDH release by HaCaT cells co-exposed to 0.5 Gy protons and 25 J/m² or 50 J/m² UVB was increased (mean effect > 1) in both UVB irradiation settings (Figure S4.2), but differences were not statistically significant between the LDH released when cells were co-exposed to protons and UVB or were challenged with single radiations. Nevertheless, the cellular systems at 48 h post-exposure were unstable regarding LDH release in independent experiments both in the case of Hs27 fibroblasts (Figure S3.3) and HaCaT keratinocytes (Figure S4.2), limiting the significance of results.”

Regarding the 2nd Major correction “Figure 2b - These LDH assay columns have extraordinarily large error bars. It is understood that some types of assays by their nature have more variability and higher standard deviations than other assays, but these are larger than any I have ever seen published. At minimum, these error bars mean that no accurate comparisons can be made because statistical significance is impossible to establish. When researchers get results like this it is common to repeat the experiment using a larger number of samples to improve the statistics or to refine/adjust the methodology used in the assays to try to reduce dispersion or to do both of these things. It is likely that the experiment that generated the graph in Figure 2b should be repeated to produce results that can be interpreted.” we provide the following response:

Response: Indeed, cellular systems at 48 h post-exposure were unstable regarding LDH release in independent experiments both in the case of Hs27 fibroblasts (Figure S3.3.) and HaCaT keratinocytes (Figure S4.2), limiting the significance of results. Due to beamtime restrictions, there is unfortunately no possibility to make new additional experiments. For the moment, LDH release results indicate that there was no significantly increased plasma membrane alteration in co-exposed cells compared to cells exposed to single radiations. We transferred though the LDH results in the Supplementary file to reduce volume in the main text.

Regarding the 3rd Major correction “Many of the graphs have a prominent red horizontal line that represents the level observed in untreated cells. This may be okay, but it is unusual because we don't get to see the error bars on the untreated samples. We would normally ask if the results with the treated samples are higher than those of the untreated samples and if they are outside the standard deviation seen in the untreated samples. This information is not available when a simple horizontal line is used to represent the untreated controls.” we provide the following response:

Response: You are absolutely right that the error bars for the untreated control samples are important for understanding the variability and comparing the treated samples effectively. To address this, we have now included the standard deviation for the untreated control samples in the legend of each figure. This addition allows for a clearer comparison of the treated samples with the untreated controls, as it provides information on the variability within the untreated group. We hope this resolves the concern, and the revised figures can now be found with these changes.

Regarding the 4th Major correction “Figure 3b - Should 0.5 Gy have a blue column for the 1h timepoint like the others or a space for it if the value was zero?” we provide the following response:

Response: The value for the 0.5 Gy condition at the 1-hour timepoint is indeed zero, so we intentionally left an empty space in the bar to reflect this. However, we realize that, by mistake, the red line representing the control did not include this empty space. We have now corrected this and extended the red line to align properly with the zero value at the 1-hour timepoint. The revised figure now accurately reflects this adjustment.

Regarding the 5th Major correction “Figure 2a and other graphs' labels - Should p < 0.5:* be p < 0.05:*?” we provide the following response:

Response: Thank you for pointing this out. We have corrected the label from ‘p < 0.5’ to ‘p < 0.05’ in Figure 2a and all other relevant graphs. We also define only the p-values that are relevant to each specific graph to avoid redundancy.

Additionally, all minor corrections have been made and are reflected using track changes.

We appreciate your careful review and helpful comments.

Kind regards

Reviewer 2 Report

Comments and Suggestions for Authors

Thank you for considering me as a reviewer for this article. From the conclusion part, I found that this work was performed in the framework of the research project BIOSPHERE. I agree with the authors that the presented topic is of high interest because the biological effects of UV and cosmic radiation on human health have not yet been investigated due to the lack of proper methodologies and measurement infrastructure.

Even that, in my opinion, this article must be improved significantly and changed (major revision) or rejected and then re-submitted.

Here, I present my major comments. The article is written as a monograph, many sentences are obvious and not connected with the aim of the study. Definitely must be shortened. For example, the introduction part. The authors present a long introduction without gaps and scientific problems in this emerging topic. I am wondering what the is difference between the energy of protons from cosmic rays versus those used in this study - this must be elaborated. The title also should be rewritten to reflect mostly the scientific problem presented in the study.

The authors in detail presented the materials and methods part, then the results section. The research design applied here is controversial. Only two donors with applied the dicentrics method (what about individual radiosensitivity!) compared to the different cell lines . The results part is too long, and some figures are not visible (i.e. Fig. 3). The authors put Table 1 as a briefing, but still it covers 2-3 pages (in reviewed manuscript). 

Author Response

Response to comments by Reviewer 2:

We appreciate the reviewer’s insightful comments. You are absolutely right that the manuscript contained lengthy sections that were not always directly connected to the study's main aim. We have carefully revised and reduced the text to make it more concise, particularly in the introduction. We have also ensured that the introduction clearly presents the scientific gaps and problems in this emerging topic. We have shortened the manuscript by removing extraneous details (in total 3 pages Introduction part is significantly shorten along with some sections in methodology part (part of DNA Damage immunofluorescence study is transferred in Supplementary file).

Regarding the following comment “I am wondering what the is difference between the energy of protons from cosmic rays versus those used in this study - this must be elaborated” we provide this response:

Response: Cosmic ray protons at ground level exhibit a broad and relatively flat energy distribution spanning from 0.01 MeV to 100 MeV, as shown in Fig. 1 (dashed red line). Beyond 100 MeV, the proton flux drops sharply. The 10 MeV energy used in our experiments aligns with the broad peak centered around 45 MeV. While the proton flux is typically low under normal solar conditions at ground level (~10⁻⁶ cm⁻² s⁻¹ MeV⁻¹), it can increase significantly during major solar particle events or at higher latitudes. For example, the proton flux rises by over two orders of magnitude when moving from ground level to an altitude of approximately 10 km (in the range of cruising altitudes for commercial airliners). The stopping power of protons used in our experiments are comparable to that of cosmic muons with energies around 1 MeV, as illustrated in Figure 1 (black solid line). This additional information also included in the introduction part.

We agree that the title should better reflect the central scientific problem addressed in our study. We have revised it accordingly to highlight the key aspects of our research more clearly. The new title is: “Synergistic Effects of UVB and Ionizing Radiation on Human Non-Malignant Cells: Implications for Ozone Depletion and Secondary Cosmic Radiation Exposure”.

Regarding the following comment “The research design applied here is controversial. Only two donors with applied the dicentrics method (what about individual radiosensitivity!) compared to the different cell lines.” we provide this response:

Response: Dicentric chromosomes are the biomarker of choice for investigating recent exposure to IR as its natural occurrence is very low (0.5–1 dic/1000 cells scored in unexposed individuals), the in vivo and in vitro induction after radiation exposure correlates. The basic finding, justifying the application of the dicentric analysis for individual dose assessment with reference to an in vitro generated standard calibration curve (reference data obtained by analysis of only few donors) is that the number of dicentrics is accepted to be proportional to the radiation dose absorbed. Otherwise, the assay could not be used as a biodosimetric test. Therefore, we think, the analysis of two different apparently healthy donors (1 male and 1 female in the age range of 20-30 years) with regard to radiation-induced dicentric formation is meaningful. Both donors did not show a significant difference with regard to dic induction by protons (p<0,005).

We appreciate the reviewer’s comments regarding the length of the results section and the clarity of the figures. We have carefully revised and condensed the results section to improve readability and focus on the key findings. Figures regarding LDH release were transferred in Supplementary file to reduce the length of results part.

Additionally, we have adjusted Figure 3 to enhance its visibility and ensure clarity by using larger letters in titles and focusing on the key findings by defining only the mentioned asterisks in each figure.

Regarding Table 1, we understand that its length may be excessive. We have reformatted and streamlined it to make it more concise while still preserving the necessary details.

Thank you for your helpful suggestions.

Best regards

Reviewer 3 Report

Comments and Suggestions for Authors

Reviewers' comments to article entitled: Biological effects of UVB and protons or gamma rays on human non-malignant cells

The article's subject is very interesting for both the medical community and biologists involved in research exploring the response of cells and organisms to radiation. The Authors selected methods that are adequate to the goals set in this study.  Despite this, in my opinion, some issues should be addressed before publication:

  1. The Results section should include the expression levels of all genes investigated. It can be easily included in supplement files.
  2. In the Introduction section, the Authors declare the following goals of the experiment: “Our goal is to provide comprehensive insights on the responses of human normal cells under these conditions and explore the potential to extrapolate and converge toward more realistic environmental scenarios. This approach provides a window of opportunity to understand how the thresholds or trigger points identified in extreme simulations could offer predictive leverage on the destabilization of biological systems, and how feedback loops and nonlinear effects might appear or break down”. Can the Authors briefly address these goals in the Discussion or Conclusions section?
  3. The Discussion chapter can be improved. In my opinion, attention should be paid to the following elements:
  4. Results concerning individual cell lines cannot be treated as typical for all cells of a given type.
  5. Results obtained for different cell lines cannot be directly compared, because they do not come from the same person. In the human population, we observe high inter-individual variability of all metabolic pathways studied in the work.
  • Analogously to cell lines, cells obtained from donors will also show differences in the efficiency of the metabolic pathways studied. Generalized results can only be reported after researching a larger number of participants. It should be clearly stated and discussed.
  1. It should be noted that the number of independent experiment repetitions should be at least three not two. It is also the study's limitation.
  2. Issues that have not been directly proven by experiment should be treated as hypotheses and presented as such in the discussion.
  3. Minor editorial errors or inaccuracies should be corrected, e.g. the title of Figure S3.2.  in the supplement section, symbols of genes are usually written in italics.

Author Response

Response to comments by Reviewer 3:

We would like to thank the reviewer for their valuable feedback and constructive comments on our manuscript. Their insights have greatly helped improve the quality and clarity of our work.

Regarding the following comment: “The Results section should include the expression levels of all genes investigated. It can be easily included in supplement files“ we provide the following response:

Response: Due to the large number of data in the form of 2(-dCt) we could not incorporate them in the Supplementary data, but upload as non-published material.  (Excel spreadsheet: Gene expression data). 

Regarding the following comment “In the Introduction section, the Authors declare the following goals of the experiment: “Our goal is to provide comprehensive insights on the responses of human normal cells under these conditions and explore the potential to extrapolate and converge toward more realistic environmental scenarios. This approach provides a window of opportunity to understand how the thresholds or trigger points identified in extreme simulations could offer predictive leverage on the destabilization of biological systems, and how feedback loops and nonlinear effects might appear or break down”. Can the Authors briefly address these goals in the Discussion or Conclusions section?” we provide the following response:

Response: We agree that addressing these goals in the Discussion section would strengthen the manuscript. In response, we have revised the Discussion and Conclusions to explicitly reflect how our findings contribute to understanding the responses of human normal cells to combined UVB and ionizing radiation exposure. Specifically, we have highlighted how our results demonstrate increased genomic instability, persistent DNA damage, and distinct stress gene expression patterns in co-exposed cells. These findings support the hypothesis that combined radiation exposure can exacerbate cellular stress responses, providing insights into potential thresholds or trigger points that may destabilize biological systems. Furthermore, our study offers preliminary evidence on how these effects might translate to real-world environmental scenarios, such as increased radiation exposure due to ozone depletion. We added a paragraph in the Discussion section (last paragraph of the section).

Regarding the following comment “The Discussion chapter can be improved. In my opinion, attention should be paid to the following elements:

  1. Results concerning individual cell lines cannot be treated as typical for all cells of a given type.
  2. Results obtained for different cell lines cannot be directly compared, because they do not come from the same person. In the human population, we observe high inter-individual variability of all metabolic pathways studied in the work.” We provide the following response:

Response:

We appreciate the reviewer’s insightful comments and acknowledge the importance of considering inter-individual variability when interpreting our results. We fully agree that results obtained from individual cell lines cannot be generalized to all cells of a given type, as each cell line may exhibit distinct biological responses due to genetic and epigenetic differences. Additionally, we recognize that comparing results from different cell lines derived from different donors has inherent limitations, as inter-individual variability in metabolic pathways and radiation sensitivity can influence the observed effects.

In response to this comment, we have revised the Discussion section to clarify these points and explicitly acknowledge the limitations of our study regarding inter-individual variability. We have also emphasized that while our findings provide important mechanistic insights, further studies using a larger number of primary cells from different donors would be necessary to better account for natural biological variability.

Regarding the following comment “Analogously to cell lines, cells obtained from donors will also show differences in the efficiency of the metabolic pathways studied. Generalized results can only be reported after researching a larger number of participants. It should be clearly stated and discussed.

  1. It should be noted that the number of independent experiment repetitions should be at least three not two. It is also the study's limitation.
  2. Issues that have not been directly proven by experiment should be treated as hypotheses and presented as such in the discussion.” We provide the following response:

Response:

  1. We agree that conducting three independent experiments would provide even greater statistical power and robustness to the findings. However, due to experimental limitations, we performed two independent experiments while ensuring reliability through the use of 2-3 replicate cultures in each condition. This approach strengthens the reproducibility of our results. Nevertheless, we acknowledge this as a study limitation.
  2. The dicentric assay in PBMCs was conducted using samples from two donors due to organizational constraints. However, dicentric chromosomes are the biomarker of choice for assessing recent IR exposure, as their natural occurrence is very low (0.5–1 dic/1000 cells in unexposed individuals), and their in vivo and in vitro induction after radiation exposure correlates well. The fundamental principle justifying the use of dicentric analysis for individual dose assessment is that the number of dicentrics is proportional to the absorbed radiation dose, as demonstrated by in vitro calibration curves derived from a limited number of donors. Otherwise, the assay would not have been widely accepted as a valid test. In our study, both apparently healthy donors (one male and one female, aged 20–30) exhibited comparable dicentric induction by protons (p<0.005), supporting the robustness of our findings. Furthermore, the replicate cultures set up per experiment and donor (a total of four cultures per condition) enhance the reliability of our results. Nevertheless, we acknowledge that the limited number of donors may be a study limitation, as inter-individual variability in radiosensitivity could influence the observed responses.

Minor editorial revisions have been made, including the correction of the title for Figure S3.2 in the supplement section. Additionally, gene symbols have been italicized, in accordance with standard formatting conventions.

These revisions ensure that the initially stated goals are clearly addressed within the manuscript. Thank you for this valuable suggestion.

Best regards

Round 2

Reviewer 2 Report

Comments and Suggestions for Authors

The authors included my comments and suggestions. The changed title more reflects the scope of research.

Author Response

We appreciate your valuable feedback. The revised title better reflects the scope of our research, aligning more closely with the study's objectives and findings. Thank you for your insightful recommendations.

Reviewer 3 Report

Comments and Suggestions for Authors

The authors addressed my comments, which improved the readability of the manuscript and clarified the main conclusions for readers.

Author Response

Thank you for your positive feedback. We appreciate your comments, which helped enhance the readability of the manuscript and clarify our main conclusions. Your insights were valuable in improving the overall quality of the paper.